# Skill of seasonal flow forecasts at catchment scale: an assessment across South Korea

Yongshin Lee[1], Francesca Pianosi[1], Andres Peñuela[2], Miguel Angel Rico-Ramirez[1]

[1] School of Civil, Aerospace and Design Engineering, University of Bristol, Bristol, BS8 1TR, UK

[2] Department of Agronomy, Unidad de Excelencia María de Maeztu, University of Cordoba, Cordoba, 14071, Spain

**Correspondence:** Yongshin Lee (yongshin.lee@bristol.ac.uk)

**Abstract.**

*Recent advancements in numerical weather predictions have improved forecasting performance at longer lead times. Seasonal weather forecasts, providing predictions of weather variables for the next several months, have gained significant attention from researchers due to their potential benefits for water resources management. Many efforts have been made to generate Seasonal Flow Forecasts (SFFs) by combining seasonal weather forecasts and hydrological models. However, producing SFFs with good skill at a finer catchment scale remains challenging, hindering their practical application and adoption by water managers. Consequently, water management decisions, both in South Korea and numerous other countries, continue to rely on worst-case scenarios and the conventional Ensemble Streamflow Prediction (ESP) method.*

*This study investigates the potential of SFFs in South Korea at the catchment scale, examining 12 reservoir catchments of varying sizes (ranging from 59 to 6648 km2) over the last decade (2011-2020). Seasonal weather forecasts data (including precipitation, temperature and evapotranspiration) from the European Centre for Medium-Range Weather Forecasts (ECMWF system5) is used to drive the Tank model (conceptual hydrological model) for generating the flow ensemble forecasts. We assess the contribution of each weather variable to the performance of flow forecasting by isolating individual variables. In addition, we quantitatively evaluate the overall skill of SFFs, representing the probability of outperforming the benchmark (ESP), using the Continuous Ranked Probability Skill Score (CRPSS). Our results highlight that precipitation is the most important variable in determining the performance of SFFs, and temperature also plays a key role during the dry season in snow-affected catchments. Given the coarse resolution of seasonal weather forecasts, a linear scaling method to adjust the forecasts is applied, and it is found that bias correction is highly effective in enhancing the overall skill. Furthermore, bias corrected SFFs have skill with respect to ESP up to 3 months ahead, this being particularly evident during abnormally dry years. To facilitate future applications in other regions, the code developed for this analysis has been made available as an open-source Python package.*

**Keywords:** Seasonal weather forecasts, Seasonal flow forecasts, Skill assessment, Ensemble Streamflow Forecast, CRPSS, Linear scaling

## 1. Introduction

Over the last decade, numerical weather prediction systems have improved their forecasting performance at longer lead times, ranging from 1 to several months ahead (Alley et al., 2019; Bauer et al., 2015). The water management sector may benefit considerably from these advances. In particular, predictions of weather variables such as precipitation and temperature several months ahead ('seasonal weather forecasts' from now on) might be exploited to anticipate upcoming dry periods and implement management strategies for mitigating future water supply deficits (Soares and Dessai, 2016).

To increase relevance for water resource management, seasonal weather forecasts can be translated into Seasonal Flow Forecasts (SFFs) via a hydrological model. SFFs can be provided and evaluated at different temporal and spatial resolutions: a coarser resolution, e.g., magnitude of total next-month runoff over a certain region (Arnal et al., 2018; Prudhomme et al., 2017) or a finer resolution, e.g., daily/weekly flow at a particular river section over the next month (Crochemore et al., 2016; Lucatero et al., 2018). This distinction is important here because coarser resolution SFFs can only be applied to inform water management in a qualitative way, whereas finer resolution SFFs can also be used to force a water resource system model for a quantitative appraisal of different management strategies. Proof-of-principle examples of the latter approach are provided by Boucher et al. (2012), Chiew et al. (2003), and Peñuela et al. (2020). These papers have demonstrated, through model simulations, the potential of using SFFs to improve the operation of supply reservoirs (Peñuela et al., 2020), irrigation systems (Chiew et al., 2003) and hydropower systems (Boucher et al., 2012).

Obviously, generating SFFs with good skill at finer scales is challenging and the lack of forecasting performance
is often cited as a key barrier to real-world application of SFFs by water managers (Jackson-Blake et al., 2022;
Soares and Dessai, 2016; Whateley et al., 2015). In practice, if a Water Resource System (WRS) model is used to
simulate and compare different operational decisions, this is done by forcing the WRS model against a repeat of
a historical low flow event ("worst-case" scenario) (Yoe, 2019) or against the Ensemble Streamflow Prediction
(ESP). ESP is a widely used operational forecasting method whereby an ensemble of flow forecasts is generated
by forcing a hydrological model with historical meteorological observations (Baker et al., 2021; Day, 1985). Since
the hydrological model is initialised at current hydrological conditions, ESP is expected to have a certain level of
performance, particularly in 'long-memory' systems where the impact of initial conditions last over long time
periods (Li et al., 2009). Previous simulation studies that examined the use of SFFs to enhance the operation of
water resources systems (e.g., Peñuela et al., 2020, as cited above) did indeed show that ESP serves as a 'hard-to-
beat' benchmark. Similar to other countries, in South Korea, the worst-case scenario and ESP are used for
informing water management activities, whereas SFFs are not currently applied. Before the use of SFFs can be
proposed to practitioners, it is thus crucial to understand the skill of such products with respect to ESP.
Numerous studies were conducted on the skill of SFFs in different regions of the world. Some of these studies
focused on the 'theoretical skill', which is determined by comparing SFFs with pseudo-observations produced by
the same hydrological model when forced with observed temperature and precipitation. This experimental set-up
enables to isolate the contribution of the weather forecast skill to the flow forecast skill, regardless of structural
errors that may be present in the hydrological model. In general, most studies have found that the theoretical skill
of SFFs may be only marginally better than that of ESP in specific region and lead time. For example, Yoseff et
al. (2013) analysed multiple large river basins worldwide and found that SFFs generally perform worse than ESP.
Likewise, the findings of Greuell et al. (2019) indicated that SFFs are more skillful than ESP for the first lead
month only.  Across Europe, the theoretical skill of SFFs was found to be higher than ESP in coastal and
mountainous regions (Greuell et al., 2018).
Although important to how the information content of seasonal weather forecasts vary across regions with
different climatic characteristics, from a water management perspective, the theoretical skill may not be the most
appropriate metric, as it reflects the performance within the modelled environment (Pechlivanidis et al., 2020)
rather than the real-world.  The 'actual skill', which is determined by comparing SFFs to flow observations, would
be more informative for water managers to decide on whether to use SFFs and when. Previous studies that
investigated the actual skill showed that, as expected, the actual skill is lower than the theoretical skill due to
errors in the hydrological model and in the weather input observations (Greuell et al., 2018; van Dijik et al., 2013).
In addition, due to the coarse horizontal resolution of seasonal weather forecasts (around $1°\times1°$), the forecast skill
can be significantly improved through bias correction, particularly of precipitation forecasts (e.g., Crochemore et
al., 2016; Lucatero et al., 2018; Tian et al., 2018). However, even after bias correction, SFFs were found unable
to surpass ESP in many previous applications (e.g., Crochemore et al., 2016; Greuell et al., 2019; Lucatero et al.,
89  2018).
Previous studies reviewed above have mainly used the seasonal weather forecasts provided by the European
Centre for Medium-Range Weather Forecasts (ECMWF). Here, it is important to note that the majority of these
studies have utilized ECMWF's system 3 (e.g., Yossef et al., 2013) or 4 (e.g., Crochemore et al., 2016; Greuell et
al., 2019; Lucatero et al., 2018; Tian et al., 2018). A few studies comparing the performance of SFFs and ESP
have been conducted based on ECMWF's cutting-edge forecasting system 5, which became operational in
November 2017. These include Peñuela et al., 2020 and Ratri et al., 2023, which however did not analyse the skill
of SFFs in much detail but rather focused on their operational implementation. Given that the upgrade of
forecasting system can lead to substantial enhancement in the performance (e.g., Johnson et al., 2019; Köhn-Reich
and Bürger, 2019), it is interesting to assess whether improved skill of weather forecasts delivered by the System
5 translates into improved skill of flow forecasts.
Our previous research (Lee at al., 2023) on the skill of seasonal precipitation forecasts across South Korea showed
that, among various forecasting centres, ECMWF provides the most skilful seasonal precipitation forecasts,
outperforming the climatology (based on historical precipitation observations). This is particularly evident during
the wet season (June to September) and in dry years, where skill can also be high at longer lead times beyond the
first month. Given the significant correlation between precipitation and flow in the country (Ministry of land,
infrastructure, and transportation, 2016), South Korea is an interesting test bed to investigate if the skill of seasonal
precipitation forecasts is mirrored into the skill of flow forecasts at the catchment scale.
Specifically, in this study we focus on 12 catchments of various size (from 59 to 6648 km$^2$) which include the
most important multipurpose reservoirs across South Korea, and where the use of SFFs may be considered for
assisting operational decisions and mitigating impacts of droughts. Given this practical long-term goal, our study
focuses on assessing the 'overall skill', which represents the probability that SFFs outperform the benchmark

(ESP) when comparing the flow forecasts with historical flow observations. As a hydrological model, we use the lumped Tank model (Sugawara et al., 1986) which is the rainfall-runoff model currently in use for the national water management and planning. For all catchments, we briefly analyse the hydrological model performance, also investigate which weather forcing input (precipitation, temperature, and potential evapotranspiration) contributes most to the performance of SFFs across different catchments, before and after bias correction. Finally, we look at how the overall skill varies across seasons, years, and catchment, to draw conclusions on when and where SFFs may be more informative than ESP for practical water resources management. In doing so, we develop a workflow for SFFs analysis implemented in a Python Jupyter Notebook, which can be utilized by other researchers for evaluating and testing SFFs in various regions.

## 2. Material and methodology

### 2.1 Study site and data

#### 2.1.1 Study site

The spatial scope of this study is defined as 12 multi-purpose reservoir catchments across South Korea. While there are 20 multi-purpose reservoirs nationwide (K-water, 2022), we have specifically selected 12 reservoirs with at least 10 years of flow observation and no external flows from other rivers or reservoirs. The location of the catchments and the mean annual precipitation, temperature, and potential evapotranspiration (PET) are shown in Figure 1(a-c). The weather data for the selected reservoir catchments is reported in Table 1.

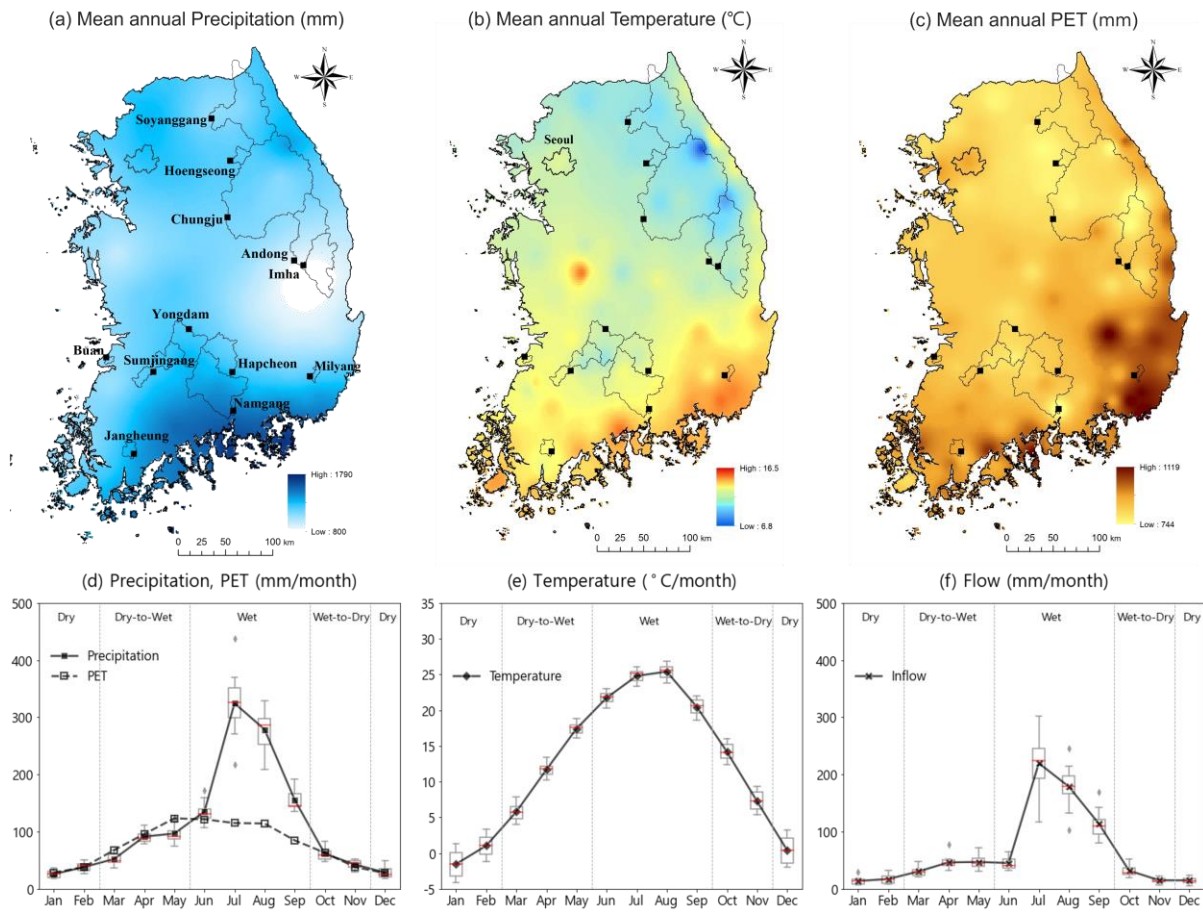

**Figure 1: Top row: mean annual (a) precipitation, (b) temperature and (c) PET across South Korea over the period 1967-2020. Black lines are the boundaries of the 12 reservoir catchments analysed in this study (all maps obtained by interpolating point measurements using the inverse distance weighting method). Bottom row: (d) cumulative monthly precipitation and PET, (e) mean monthly temperature and (f) cumulative monthly flow. These three variables are averaged over the 12 reservoir catchments from 2001 to 2020. Box plots show the inter-catchment variability.**

**Table 1. Characteristics of the 12 multipurpose reservoirs (from North to South) and the catchments they drain (K-water, 2022). Tmin and Tmax represent mean monthly minimum and maximum temperature averaged over 2001-2020,**

**all other meteorological variables (P: precipitation, T: temperature, PET: potential evapotranspiration) are annual averages over the same period.**

| | Catchment | Soyanggang | Hoengseong | Chungju | Andong | Imha | Yongdam | Buan | Sumjingang | Hapcheon | Milyang | Namgang | Jangheung |
|---|---|---|---|---|---|---|---|---|---|---|---|---|---|
| Mean annual | Area (km²) | 2703 | 209 | 6648 | 1584 | 1361 | 930 | 59 | 763 | 925 | 95 | 2285 | 193 |
| | P (mm) | 1220 | 1336 | 1197 | 1079 | 956 | 1317 | 1292 | 1343 | 1279 | 1375 | 1477 | 1439 |
| | T (℃) | 10.8 | 10.9 | 11.1 | 11.1 | 12.2 | 11.8 | 13.5 | 12.6 | 12.8 | 14.2 | 13.5 | 13.8 |
| | T min | -4.2 (Jan.) | -4.0 (Jan.) | -3.2 (Jan.) | -3.5 (Jan.) | -1.6 (Jan.) | -2.3 (Jan.) | -0.1 (Jan.) | -1.5 (Jan.) | -0.8 (Jan.) | 1.0 (Jan.) | 0.4 (Jan.) | 1.3 (Jan.) |
| | T max | 24.0 (Aug.) | 24.1 (Aug.) | 25.9 (Aug.) | 23.8 (Aug.) | 25.1 (Aug.) | 24.8 (Aug.) | 26.7 (Aug.) | 25.8 (Aug.) | 25.5 (Aug.) | 26.8 (Aug.) | 26.0 (Aug.) | 26.2 (Aug.) |
| | PET (mm) | 874 | 870 | 881 | 896 | 947 | 884 | 960 | 919 | 933 | 993 | 952 | 896 |

Figure 1(d-f) shows the monthly changes in precipitation and PET (d), temperature (e) and flow (e) averaged over the 12 selected catchments from 2001 to 2020. Generally, the catchments located in the Southern region exhibit higher mean annual precipitation, temperature, and PET. In order to examine how the skill of seasonal weather and flow forecasts varies across a year, we divide a year into four seasons based on monthly precipitation (Lee et al., 2023): dry (December to February), dry-to-wet transition (March to May), wet (June to September), wet-to-dry transition (October to November). As shown in this figure, most of the total annual precipitation (and the corresponding flow) occurs during the hot and humid wet season, while the dry season is characterized by cold and dry conditions. The high inter-annual variability of precipitation and flow is a feature of South Korea's climate and is attributed to the impacts of typhoons and monsoons (Lee et al., 2023). Figure 1(d-f) also shows high inter-catchment variability during the wet season in both precipitation (d) and flow (f), whereas the inter-catchment variability in temperature (e) is more obvious during the dry season.

### 2.1.2 Hydrologic data and seasonal weather forecasts

Precipitation, temperature, and potential evapotranspiration are the key variables required to simulate flows using a hydrological model. To this end, daily precipitation data from 1318 in-situ stations produced by the Ministry of Environment, the Korea Meteorologic Administration (KMA), and the national water resources agency (K-water) (Ministry of Environment, 2021), and daily temperature data from 683 in-situ stations generated by KMA were obtained. Both precipitation and temperature data cover the period from 1967 to 2020 (see Figure 1). Potential evapotranspiration (PET) data was computed using the standardized Penman-Monteith method suggested by UN Food and Agriculture Organization. The precipitation and temperature measurements have been quality-controlled by the Ministry of Environment. We used the Thiessen polygon method to calculate the catchment average precipitation and temperature.

The flow data used in this study refers to the flow into the reservoir from their upstream catchment (see Table1 and Figure 1). K-water generates daily flow data through a water balance equation, which takes into account the daily changes in reservoir volume (from storage-elevation curve) caused by the water level fluctuations and water supplies. However, to date, reservoir evaporation has not been considered in the flow estimation process. In this study, quality-controlled daily flow data for each reservoir produced by K-water is used.

Several weather forecasting centres, including ECMWF, the UK Met Office and the German Weather Service, provide seasonal weather forecasts datasets through the Copernicus Climate Data Store (CCDS). According to our previous study (Lee et al., 2023), ECMWF was found to be the most skilful provider of seasonal precipitation forecasts for South Korea. Since the precipitation is one of the most important weather forcings in hydrological forecasting (Kolachian and Saghafian, 2019), we have utilized the seasonal weather forecasts datasets from ECMWF System 5 (Johnson et al., 2019) in this study. Since 1993, ECMWF has been providing 51 ensemble forecasts (a set of multiple forecasts equally likely) on a monthly basis (25 ensembles prior to 2017) with a horizontal resolution of 1° × 1° and daily temporal resolution up to 7 months ahead. In this study, the time period from 1993 to 2020 was selected and the ensemble forecasts for the selected catchments have been downloaded from the CCDS. Here, we utilized data from 1993 to 2010 to generate bias correction factors, and data from 2011 to 2020 to assess the skill (see Figure S1 in the supplementary material).

### 2.2 Methodology

The methodology of our analysis is summarized in the schematic diagram shown in Figure 2. Firstly, we compiled seasonal weather forecasts ensemble from ECMWF for precipitation (P), temperature (T), and PET over the 12 reservoirs for 10 years from 2011 to 2020. To downscale the datasets, a linear scaling method was applied to each weather forcing (Sec. 2.2.1). Secondly, we estimated the parameters of the hydrological model and validated its

performance (Sec. 2.2.2). Utilizing the seasonal weather forecasts dataset as input data to the hydrological model,
we generated an ensemble of SFFs, and using historical weather observations as input, we produced ESP.
Specifically, to calculate ESP, 45 ensemble members of each weather variable were also selected from historical
observations (1966-2010, see Figure S1). Each ensemble member represents the simulated flow using a
hydrological model initialized with observed meteorological data to simulate current conditions and forced by
historical meteorological observations for the forecasting period. The Continuous Ranked Probability Score
(CRPS) and the Continuous Ranked Probability Skill Score (CRPSS) were applied (Sec. 2.2.3) to calculate the
absolute performance (score) of each forecast product (Sec. 3.1 and 3.2) and the relative performance (overall
skill) of SFFs with respect to ESP (Sec. 3.3, 3.4).

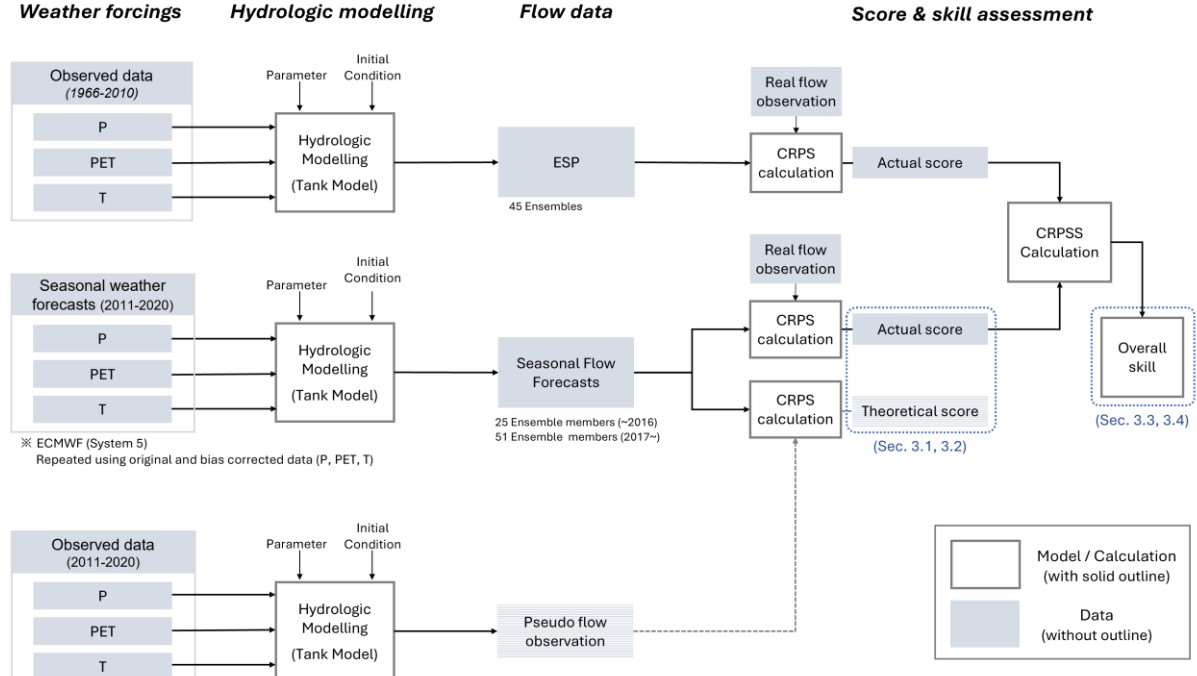


**Figure 2: Schematic diagram illustrating analysis method of the study.**
Specifically, in Section 3.1, we analysed the contribution of hydrological modelling uncertainty to the
performance of SFFs by comparing the actual score to the theoretical score, which is calculated using pseudo flow
observations. Here, pseudo-observation refers to the flow time-series obtained by feeding the hydrological model
with weather observations, i.e. where errors due to hydrological model structure are removed. In Section 3.2 we
investigated which weather variable mostly influence the performance of SFFs. For doing so, we first calculated
the 'isolated score' of the flow forecasts generated by forcing the hydrological model with seasonal weather
forecasts for one meteorological variable while using observational data for the other two variables. For instance,
to assess the contribution of precipitation, we calculated the isolated score-P using seasonal precipitation forecasts,
and observations for temperature and PET. Then, we computed the 'integrated score' using seasonal weather
forecasts for all three variables and determined the 'relative scores' for each variable as the ratio of the isolated
score over the integrated score. This workflow is illustrated in Figure S2 (supplementary material). In Sections
3.3 to 3.5, we examined the regional and seasonal variations and the characteristics of overall skill under extreme
climate conditions.

### 2.2.1 Bias correction (Statistical downscaling)

The seasonal weather forecasts datasets from CCDS have a spatial resolution of 1°×1°, which is too coarse for the
catchment-scale analysis. Previous studies also have reported that seasonal weather forecasts generated from
General Circulation Models contain systematic biases and this can cause forecast uncertainty (Manzanas et al.,
2017; Maraun, 2016; Tian et al., 2018). Moreover, the usefulness of bias correction in enhancing the forecast skill
has been shown in many previous studies (Crochemore et al., 2016; Ferreira et al., 2022, Pechlivanidis et al., 2020;
Tian et al., 2018). Hence, it is imperative to investigate the potential enhancement in the skill of hydrological
forecasts resulting from the bias correction of weather forcings.
Numerous bias correction methods have been developed including linear scaling method, local intensity scaling
and quantile mapping (Fang et al., 2015; Shrestha et al., 2017). Thanks to its simplicity and low computation cost
(Melesse et al., 2019), the linear scaling method is widely adopted. Despite its simplicity, this method has
demonstrated practical usefulness in various studies (Azman et al., 2022; Crochemore et al., 2016; Shrestha et al.,
2017), including our previous study on seasonal precipitation forecasts across South Korea (Lee et al., 2023).
Therefore, the linear scaling method was utilized in this study.
Previous studies found that additive correction is preferable for temperature whereas multiplicative correction is
preferable for variables such as precipitation, evapotranspiration, and solar radiation (Shrestha et al., 2016).
Consequently, the equations for linear scaling method for each variable can be expressed as:
$$P^*_{forecasted} = P_{forecasted} \cdot (b_P)_m = P_{forecasted} \cdot \left[ \frac{\mu_m(P_{observed})}{\mu_m(P_{forecasted})} \right] \tag{1}$$
$$PET^*_{forecasted} = PET_{forecasted} \cdot (b_{PET})_m = PET_{forecasted} \cdot \left[ \frac{\mu_m(pET_{observed})}{\mu_m(pET_{forecasted})} \right] \tag{2}$$
$$T^*_{forecasted} = T_{forecasted} + (b_T)_m = T_{forecasted} + \left[ \mu_m(T_{observed}) - \mu_m(T_{forecasted}) \right] \tag{3}$$
where $X^*_{forecasted}$ is the bias corrected forecast variable ($X$) at daily time scale, $Y_{forecasted}$ is the original forecast
variable ($Y$) before bias correction, $(b_Y)_m$ is the bias correction factors for each variable at month $m$. $\mu_m$ represents
monthly mean, and $Y_{observed}$ is the observed daily data for the variable ($Y$). In this study, daily precipitation
forecasts were bias corrected using the monthly bias correction factor ($b_m$) for each month ($m$ = 1 to 12). The bias
correction factor was computed using the observations and original forecasts datasets from 1993 to 2010, and
these were then applied to adjust each seasonal weather forecasts for later years (2011 to 2020).

### 2.2.2 Hydrologic modelling

The Tank model was first developed by Sugawara of Japan in 1961 (Sugawara et al., 1986; Sugawara, 1995) and
has become a widely used conceptual hydrologic model in many countries (Goodarzi et al., 2020; Ou et al. 2017).
A modified version of the Tank model, incorporating soil moisture structure and snowmelt modules, is commonly
used in South Korea for long-term water resources planning and management purposes due to its higher
performance (Kang et al., 2004; Lee et al., 2020). As shown in Figure 3, the modified Tank model used in this
study comprises four storage tanks representing the runoff and baseflow in the target catchment (Phuong et al.,
2018; Shin et al., 2010) and incorporates a water-balance module suggested by the United States Geological
Survey (McCabe and Markstrom, 2007).

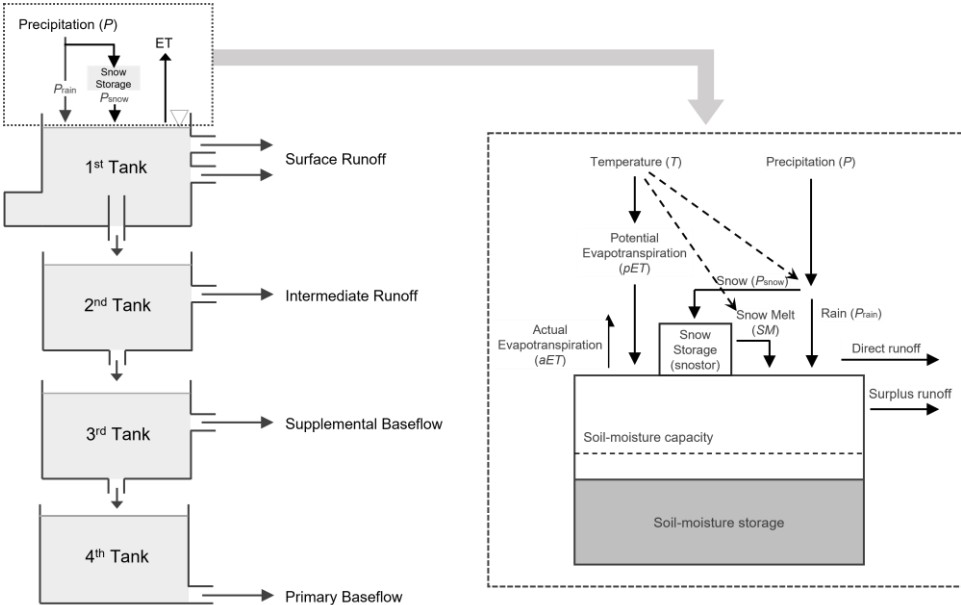


**Figure 3: The structure of modified Tank model (Left) and its water -balance module (Right)**

This model has 21 parameters (see Table S1 in the supplementary material), which were calibrated based on
historical observations. We calibrated the model using observations for the period from 2001 to 2010, and the
validation was done using the time period 2011 to 2020. To estimate the model parameters, the Shuffled Complex
Evolution global optimization algorithm (SCE-UA), developed at the University of Arizona (Duan et al., 1992,
1994), is utilized. This algorithm has widely been used for the calibration of hydrologic models and has shown
more robust and efficient performance compared to many traditional optimization methods such as Genetic
Algorithm, Differential Evolution, and Simulated Annealing (Rahnamay-Naeini et al., 2019; Yapo et al., 1996).
The following Objective Function (OF) proposed by Sugawara (Sugawara et al., 1986), is applied for the SCE-
UA algorithm, because a previous study demonstrated that this objective function shows superior results in
calibrating the Tank model in South Korean catchments with calibration periods longer than 5 years (Kang et al.,
251 2004).

$$OF = \sum_{t=1}^{N} \left| q_t^{obs} - q_t^{sim} \right| / q_t^{obs} \tag{4}$$

where $t$, $N$ represent time (in days) and total number of time steps, $q_t^{obs}$ and $q_t^{sim}$ represent the observed and
simulated flow at time $t$, respectively. The optimal parameter set is the one that produces the lowest value from
the objective function.
In order to evaluate the model performance in diverse perspectives, we used three different evaluation indicators:
Nash-Sutcliffe model Efficiency coefficient (*NSE*), Percentage Bias (*PBIAS*), and Ratio of Volume (*ROV*). The
calculation of each indicator was carried out as described by the following equations.

$$NSE = 1 - \sum_{t=1}^{N}(q_t^{obs} - q_t^{sim})^2 / \sum_{t=1}^{N}(q_t^{obs} - q_{mean}^{obs})^2 \tag{5}$$

$$PBIAS = \sum_{t=1}^{N}(q_t^{obs} - q_t^{sim})^2 / \sum_{t=1}^{N} q_t^{obs} \times 100 \tag{6}$$

$$ROV = \sum_{t=1}^{N} q_t^{sim} / \sum_{t=1}^{N} q_t^{obs} \tag{7}$$

where $t$, $N$, $q_t^{obs}$ and $q_t^{sim}$ are as defined in Eq.4, and $q_{mean}^{obs}$ represents observed mean flow across the total
number of time steps ($N$).
The *NSE* can range from negative infinity to 1. A value of 1 indicates a perfect correspondence between the
simulated and the observed flow. *NSE* values between zero and 1 are generally considered acceptable levels of
performance (Moriasi et al., 2007). *PBIAS* is a metric used to measure the average deviation of the simulated
values from the observation data. The optimal value of *PBIAS* is 0, and low-magnitude values indicate accurate
simulation. Positive (negative) values of *PBIAS* indicate a tendency for overestimation (underestimation) in the
hydrologic modelling (Gupta et al., 1999). *ROV* represents the ratio of total volume between the simulated and
observed flow. An optimal *ROV* value is 1, and a value greater (less) than 1 suggests overestimation
(underestimation) of total flow volume (Kang et al., 2004).

### 2.2.3 Score and skill assessment

As a score metric, we adopted the CRPS developed by Matheson and Winkler (1976) which measures the
difference between the cumulative distribution function of the forecast ensemble and the observations. The CRPS
has the advantage of being sensitive to the entire range of the forecast and being clearly interpretable, as it is equal
to the Mean Absolute Error for a deterministic forecast (Hersbach, 2000). For these reasons, it is a widely used
metric to assess the performance of ensemble forecasts (Leutbecher and Haiden, 2020). The CRPS can be
calculated as:

$$CRPS = \int [F(x) - H(x \geq y)]^2 \, dx \tag{8}$$

where $F(x)$ represents the cumulative distribution of SFFs ensemble, $x$ and $y$ are respectively the forecasted and
observed flow, $H$ is called the indicator function and is equal to 1 when $x \geq y$ and 0 when $x < y$. If SFFs were
perfect, i.e., all the ensemble members exactly matched the observations, the CRPS would be equal to 0.
Conversely, a higher CRPS indicates a lower the performance, as it implies that the forecast distribution is further
from the observation. Note that the CRPS measures the absolute performance (score) of forecast without
comparing it to a benchmark.
Along with the CRPS, we also employed the CRPSS, which presents the forecast performance in a relative manner
by comparing it to a benchmark forecasting method. It is defined as the ratio of the forecast and benchmark score
and is expressed as follows:
$$\text{CRPSS} = 1 - \frac{\text{CRPS}^{Sys}}{\text{CRPS}^{Ben}} \qquad (9)$$
where $\text{CRPS}^{Sys}$ is the CRPS of the forecasting system (SFFs in our case) and $\text{CRPS}^{Ben}$ is the CRPS of the
benchmark. The values of CRPSS can range from $-\infty$ to 1. A CRPSS value between 0 to 1 indicates that the
forecasting system has skill with respect to the benchmark. Conversely, when the CRPSS is negative, i.e., from -
$\infty$ to 0, the system gives lower performances than the benchmark. Here, we utilized ESP as a benchmark due to
its extensive application in flow forecasting (Pappenberger et al., 2015; Peñuela et al., 2020) and its computational
efficiency (Baker et al., 2021; Harrigan et al., 2018). ESP is generated using the Tank model fed with historical
daily meteorological records from 1966 to 2010. As this period covers 45 years, ESP is composed of 45 members
for each catchment.
Since the CRPSS ranges from $-\infty$ to 1, simply averaging the CRPSS values over a period can result in low or no
skills due to the presence few extremely negative values. To address this issue, here we employed the 'overall
skill' metric introduced by Lee et al. (2023). The overall skill represents the probability with which a forecasting
system (in our case, the SFFs) outperforms the benchmark (i.e., has CRPSS greater than 0) over a specific period.
It is calculated as:
$$\text{Overall skill (\%)} = \frac{\sum_{y=1}^{N_y} [\, H\,(\text{CRPSS})\,(y)\,]}{N_y} \times 100 \ (\%) \qquad (10)$$
where $N_y$ is the total number of years, the indicator function H is equal to 1 when CRPSS (y) > 0 (SFFs have skill
with respect to ESP in year y) and 0 when CRPSS (y) $\leq$ 0 (ESP outperforms SFFs). If the overall skill is greater
than 50%, we can conclude that SFFs generally have skill over ESP across the period.
**3. Results**
**3.1 Contribution of hydrological model to the performance of SFFs**

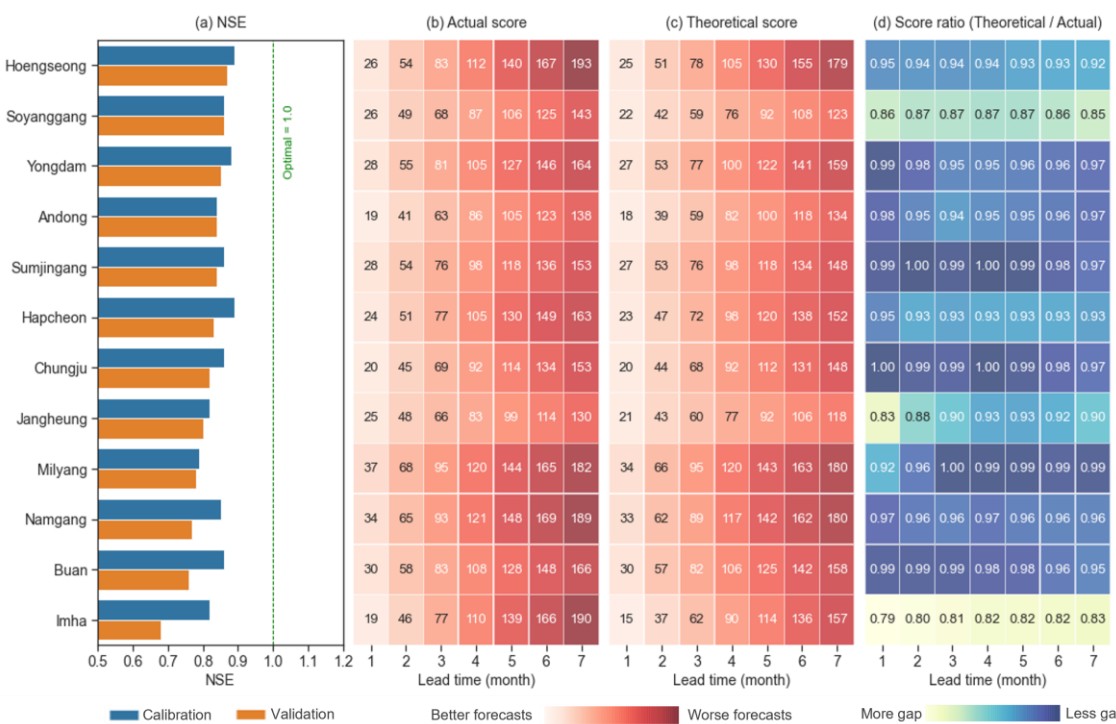

**Figure 4. (a) Nash-Sutcliffe Efficiency (NSE) of the hydrological models for the 12 catchments analysed in this study;**
**(b) actual score and (c) theoretical score of SFFs, (d) score ratio (theoretical / actual) in terms of mean CRPS at different**
**lead times (x-axis) (the scores are calculated before the bias correction of weather forcings). The actual score is**
**determined by comparing SFFs to flow observations. The theoretical score is determined by comparing SFFs to pseudo-**
**observations produced by the same hydrological model forced with observed precipitation, temperature and PET.**
Figure 4(a) shows the NSE of the modified Tank model for each catchment during the calibration period 2001-
2010 (blue bars) and the validation period 2011-2020 (orange bars). As seen in this figure, the NSE values for the
12 catchments are generally high (within the range of 0.7 to 0.9) during both the calibration and validation periods,
and the relative difference in performance between the two periods is small for all catchments. Specifically, the
NSE results indicate a 'good' performance through comparative analysis (Chiew and Mcmahon, 1993; Moriasi et
al., 2015). However, the last three catchments (Namgang, Buan and Imha) exhibit a relatively greater gap between
calibration and validation periods. Among all 12 catchments, these three exhibit the most distinctive hydrological
characteristics: Imha is the driest, while Namgang is the wettest catchment, and Buan is located along the coast,
with the smallest catchment area. A detailed model performance evaluation, including other metrics such as
PBIAS and ROV (refer to Figure S3 in the supplementary material), also supports this result. Overall, Figure 4
demonstrates that the Tank model utilized in this study shows an excellent performance in simulating flow, with
relatively higher modelling challenges observed in those three catchments.
Figures 4(b-c) represent the actual and theoretical scores (mean CRPS) over the period 2011-2020. Again, these
are calculated by comparing the simulated flows with the observed flows (actual score), and with pseudo-
observations (theoretical score), respectively. Since the CRPS is computed based on accumulated monthly mean
flow at a given lead time, forecast errors also accumulate over time. Therefore, both scores deteriorate
considerably as the lead time increases. Generally, the theoretical scores are slightly smaller than the actual scores,
but the difference is marginal.
To facilitate comparison, the ratio between the actual score and theoretical score is shown in Figure 4(d). For most
catchments, the ratio values are close to 1, confirming the small gap between actual and theoretical score. The
noticeable exception is only seen in Imha catchment, characterised by being the direst among the catchments and
exhibiting the lowest modelling performance (Figure 4(a)).

## 3.2 Contribution of weather forecasting to the performance of SFFs

In this section, we quantify the contribution of each weather forcing forecast to the performance of SFFs, as
measured by the CRPS (see Section 2.2 and Figure S2 in the supplement material for details on the underpinning
methodology). Figure 5 shows the relative scores for each non-bias corrected weather forecast across all seasons
(a), dry season (b) and wet season (c) at different lead times (1, 3, and 6 months). The relative score is calculated
as the ratio of the integrated score (computed using seasonal weather forecasts for all weather forcings), to the
isolated score (when SFFs are computed using seasonal forecasts for one weather forcing, and observations for
the other two). The closer the isolated score to the integrated score, the larger the contribution of that weather
forcing to the overall performance (or lack of performance) of the SFFs.
As shown in Figure 5(a), the contribution of each weather forecast to the performance of SFFs varies with
catchment and lead time, but overall precipitation forecast plays a dominant role. Specifically, the contribution of
precipitation forecast (red) accounts for almost 90% of the integrated score which forced by seasonal weather
forecasts for all weather forcings, while PET (orange) and temperature (blue) contribute a similar level, ranging
between 30% and 40%.
During the dry season (Figure 5(b)) however, PET and temperature show comparable levels of contribution to
precipitation. This is more evident in the Soyanggang and Hoengseong catchments, which are both located in the
northernmost region of South Korea (see Figure 1). These catchments are characterized by low temperatures and
heavy snowfall in the dry (winter) season. Correct prediction of temperature is thus crucial here as temperature
controls the partitioning of precipitation into rain and snow, and hence the generation of a fast or delayed flow
response. Further analysis (shown in the supplementary material, Figure S4), reveals that temperature forecasts in
these two catchments are consistently lower than observation, which means that the hydrological model classifies
rain as snow for several events, and hence retains that 'snow' in the simulated snowpack which in reality should
produce a flow response. This explains the significant increase in performance when forcing the model with bias
corrected temperature instead (Figure S4(b)).

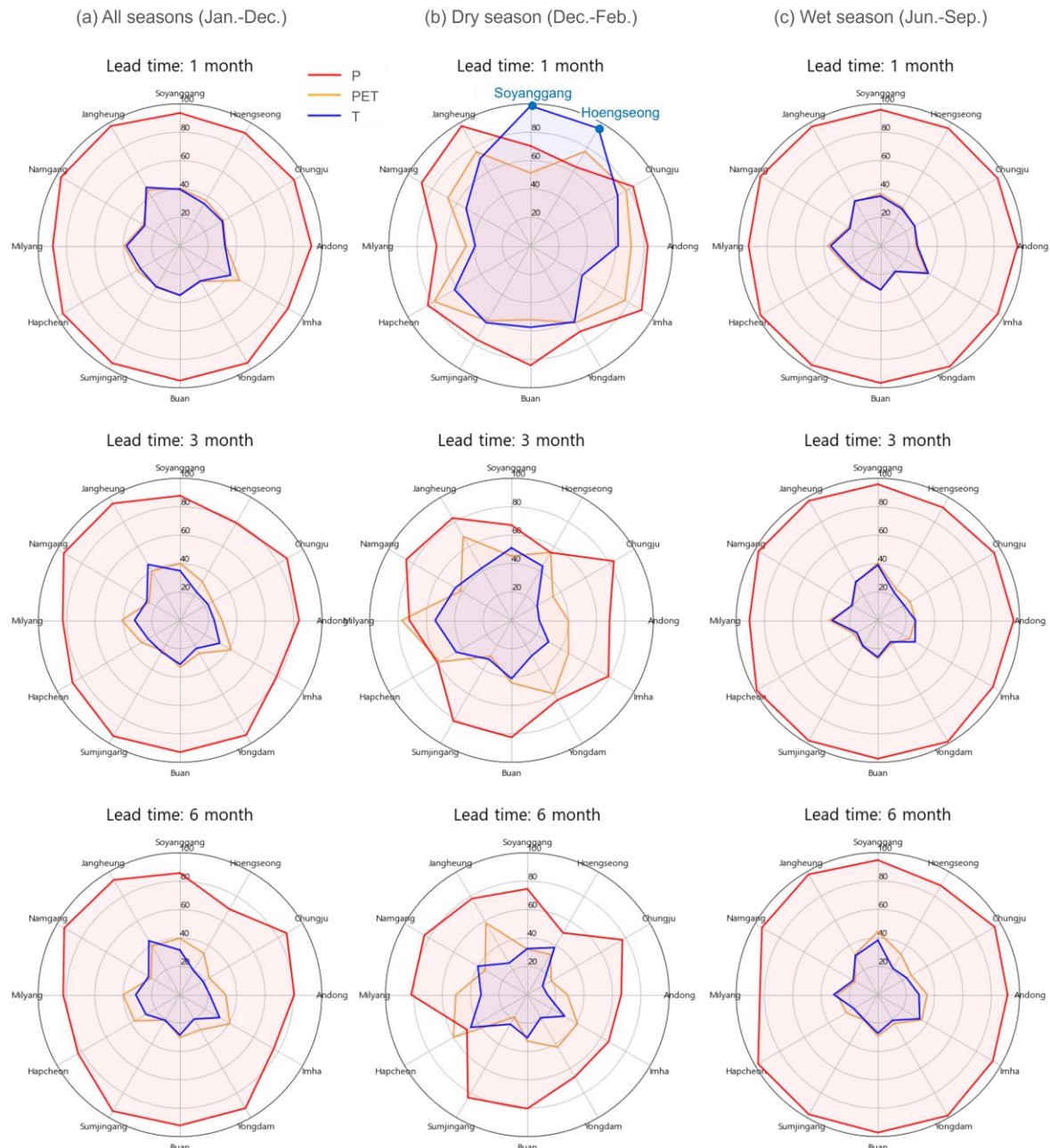

**Figure 5. Relative score (%) of each weather forcings (Precipitation: red, PET: orange, Temperature: blue) before bias correction to the score of SFFs averaged over 10 years (2011-2020) during (a) all seasons, (b) dry and (c) wet season at 1, 3 and 6 lead months from the top to bottom (Catchments are ordered by their location from the northernmost (Soyanggang) to the southernmost (Jangheung) in right-angle direction, see Figure 1).**

In order to enhance the forecasting performance, we applied bias correction to each weather forcing and re-generated SFFs with bias-corrected weather forcings. In most catchments and lead times, the overall skill is improved after correcting biases. The overall skill increases by 46% to 54% on average across all seasons, and more specifically from 31% to 50% in the dry season and from 54% to 55% in the wet season. The largest increase in overall skill is found in the Imha catchment, which had the lowest skill before correcting biases. For a detailed account of overall skill before and after bias correction, see Figure S5 and S6 in the supplementary material.

Figure 6 illustrates the change in the relative score of each weather forcing after bias correction, focusing on the dry season and the first forecasting lead month. One notable finding is that, in the snow-affected catchments (Soyanggang and Hoengseong), there is a significant decrease in the relative score of temperature after applying bias correction. As shown in detail in Figure S4 in the supplementary material, this is due to the correction of systematic underestimation biases in temperature forecasts, which lead to a more correct partition of precipitation

377 into snow and rain, and thus better flow predictions. The relative score of the forecasts for all seasons and lead
378 times after bias correction is reported in Figure S7 in the supplementary material.

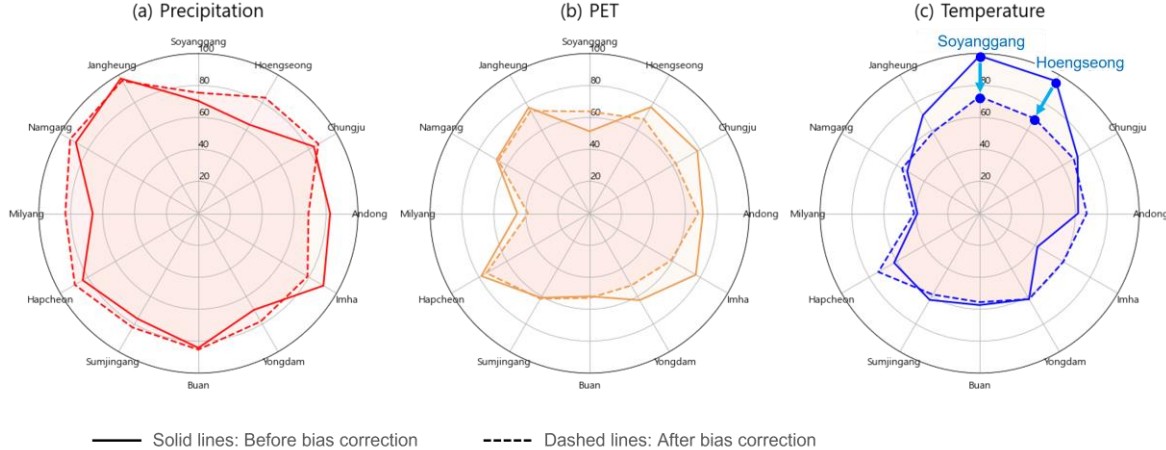

379

**Figure 6. Relative score (%) of each weather forcings ((a) Precipitation, (b) PET, (c) Temperature), before (solid line)**
**and after (dashed line) bias correction, to the score of SFFs averaged over 10 years (2011-2020) during the dry season**
**and first lead month.**

### 3.3 Comparison between SFFs and ESP across seasons and catchments

In order to comprehensively compare the performance of SFFs and ESP, we employed the overall skill, which
quantifies the frequency with which SFFs outperform ESP, as outlined in section 2.2.3 (Eq.10). Figure 7 shows
the seasonal and regional variations of overall skill (after bias correction) for all seasons (a), for the dry season (b)
and the wet season (c). For each catchment, the results are visualised through a table showing the overall skill at
lead times of 1 to 6 months. The table cells are coloured in green (pink) when SFFs outperform ESP (ESP
outperforms SFFs). Yellow colour indicates that the system and benchmark have equivalent performance. In
principle, this happens when the overall skill is equal to 50%, however in order to avoid misinterpreting small
differences in overall skill, we classified all cases as equivalent when it is between 45% and 55%. While the choice
of the range (±5%) is subjective, we find it helpful to assist analysts in avoiding spurious precision in a simple
and intuitive manner.

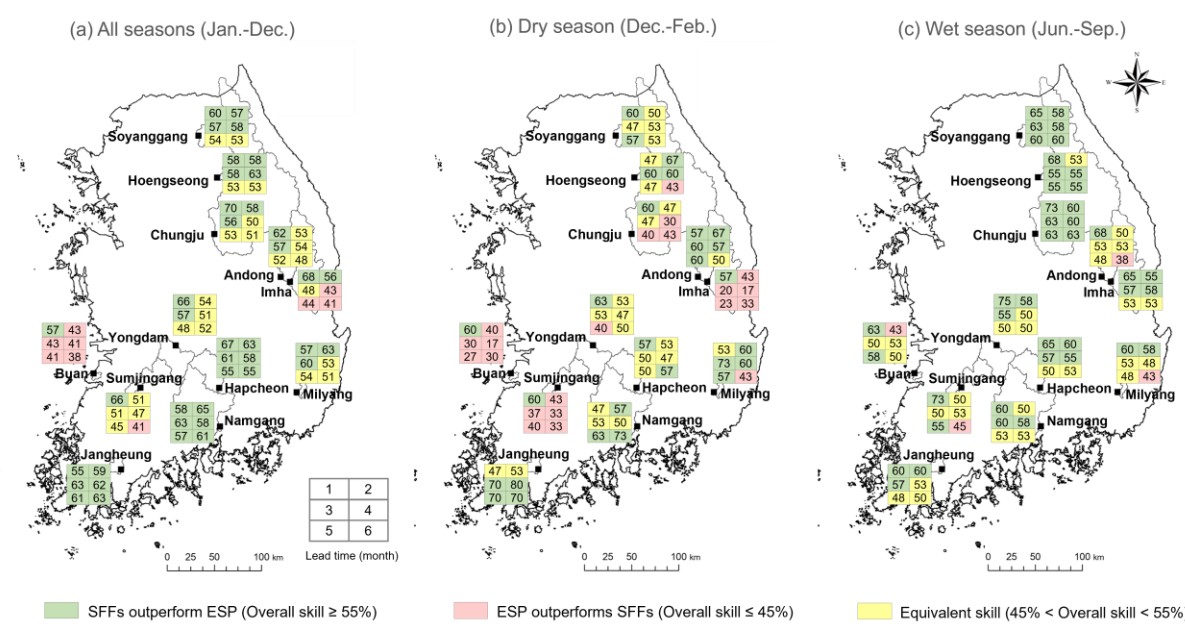


**Figure 7. Map of the overall skill of bias corrected SFFs for 10 years (2011-2020) over (a) all seasons, (b) dry season**
**and (c) wet season. The colors represent whether SFFs outperform EPS or not for each catchment and lead time (1 to**
**6 months).**

As shown in Figure 7(a), the overall skill of SFFs varies according to the lead time, season and catchment. SFFs
generally outperform ESP, particularly up to 3 months ahead. At longer lead times, the results vary from catchment
to catchment. For instance, in some catchments generally located in the Southern region, such as Janheung,
Namgang, and Hapcheon, SFFs outperform ESP for longer lead times. On the other hand, in some catchments,
such as Imha and Buan, ESP generally exhibits higher performance than SFFs. In specific, two catchments, Buan,
which is located in the Western coastal region and has the smallest catchment area, and Imha, which is the driest
catchment, show the lowest skill. Nevertheless, we could not identify a conclusive correlation between catchment
characteristics such as size or mean annual precipitation and overall skill.
Comparing the results for the dry and wet seasons, Figure 7(b-c) shows that SFFs are much more likely to
outperform ESP in the wet season, and particularly in the catchments in northernmost region. During the dry
season, overall skill of SFFs is lower, and particularly in the Buan, Imha and Sumjingang catchments SFFs
outperform ESP only for the first lead month.
**3.4 Comparison between SFFs and ESP in dry and wet years**
We now assess the influence of exceptionally dry and wet conditions on the overall skill of SFFs. Based on the
mean annual precipitation across 12 catchments within the period 2011-2020, we classified the years 2015 and
2017 as dry (P < 900 mm), and the years 2011 and 2020 as wet (P > 1500 mm). Figure 8 shows the overall skill
of SFFs averaged over 12 catchments for the entire period (a), dry years (b), and wet years (c), during all seasons
(black solid line), dry (red dashed line) and wet (blue dashed line) seasons, respectively.

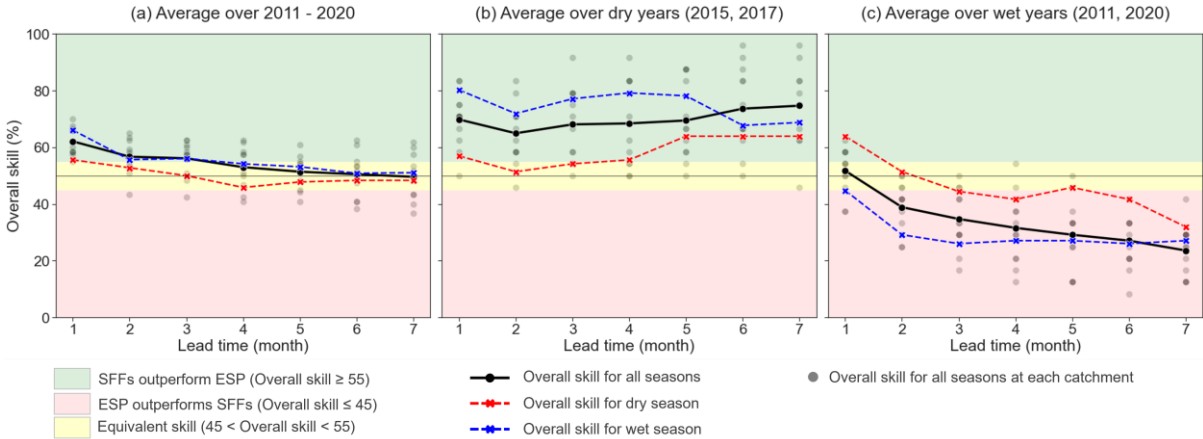


**Figure 8. Overall skill of bias corrected SFFs over 12 catchments averaged over (a) all years (2011 to 2020), (b) dry**
**years (mean annual P < 900mm) and (c) wet years (mean annual P > 1500mm) during all seasons (black lines), dry**
**seasons (red dashed lines) and wet seasons (blue dashed lines). The pale black points represent the overall skill for all**
**seasons at each catchment. Here, mean annual precipitation is averaged across the catchments and years.**
Figure 8(a) shows that SFFs outperform ESP for lead times of up to 3 months across seasons, while maintaining
equivalent performance levels thereafter. In addition, it is evident that SFFs is more skilful during the wet season
than during the dry season. In dry years (Figure 8(b)), in contrast to the typical decrease in the overall skill with
lead time, we find that SFFs maintain a significantly higher skill at all lead times, and particularly during the wet
season (blue line). On the other hand, in wet years (Figure 8(c)), the overall skill is generally poor, and ESP
generally has higher performance than SFFs, especially during the wet season.
Last, we analyse the spatial variability of the overall skill by looking at the spread of individual catchments (grey
dots). We see that the spread in dry and wet years (Figure 8(b-c)) is larger than in all years (Figure 8(a)). This
confirms that under extreme weather conditions, the uncertainty and variability in the forecasting performance
increase depending on the catchment. A more detailed analysis of the overall skill for each catchment (described
in Figure S8 in the supplementary material) shows that the catchments located in Southern region consistently
exhibit higher skill, regardless of lead times and dry/wet years.
**3.5 Example of flow forecasts time-series**
Figure 9 shows an example of the flow into the Chungju reservoir, which holds the largest storage capacity in
South Korea. The overall skill of this catchment is the highest for a 1-month lead time; however, from the second
lead month onward, it shows a moderate level of overall skill compared to other catchments (see Figure S8 in the
supplementary material). In this section, we compare the observed and forecasted cumulative flow forced by
seasonal weather forecasts (SFFs, green lines) and historical weather records (ESP, pink lines) for lead times of
1, 3, and 6 months from April during the wettest (2011) and the driest year (2015), respectively.

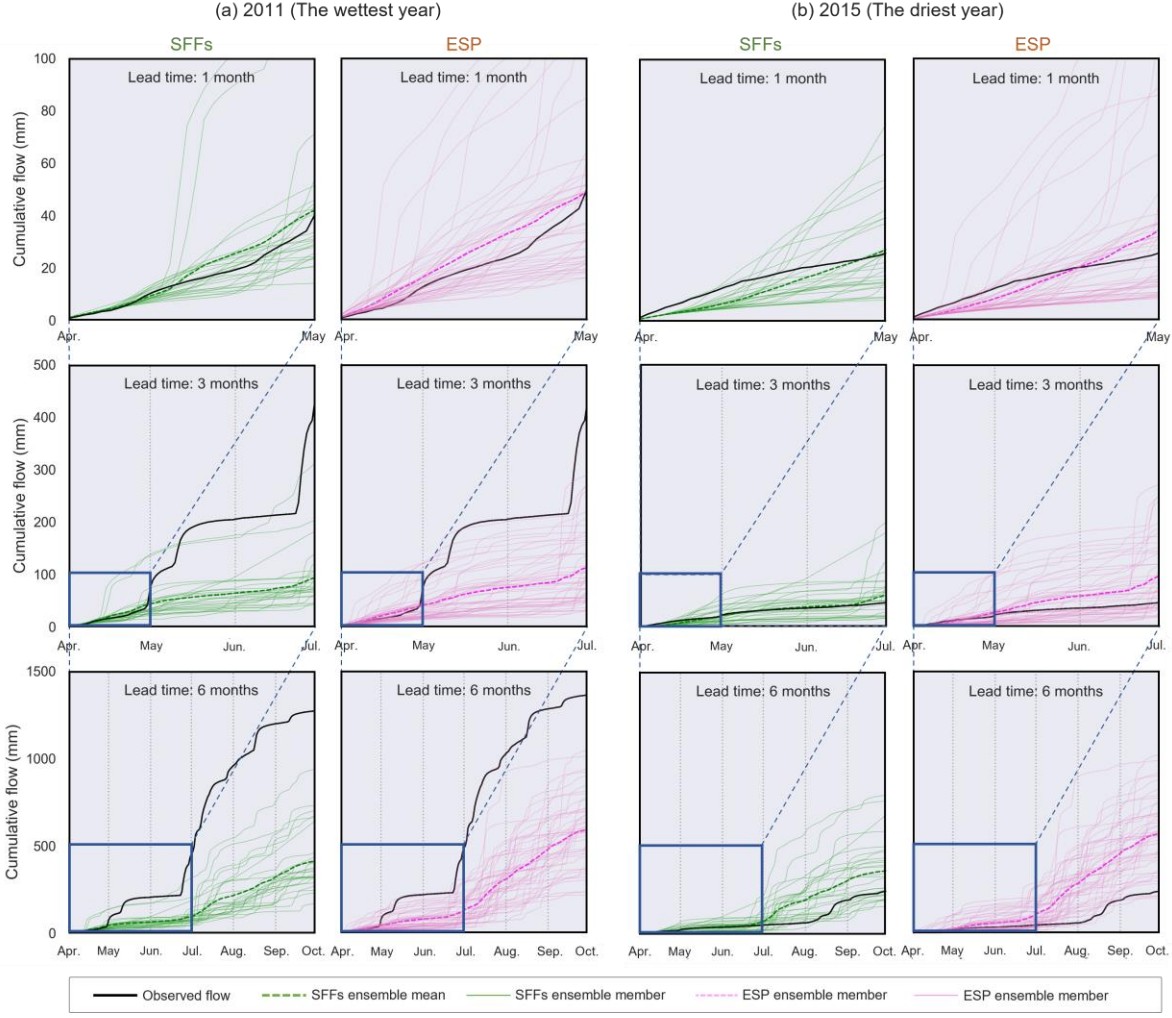

**Figure 9. Observed cumulative flow (black lines) and forecasted cumulative flow representing SFFs after**
**bias correction (left, green lines) and ESP (right, pink lines) in the Chungju reservoir for 1, 3, and 6 months**
**of lead times over (a) the wettest year (2011, 1884mm/year), and (b) the driest year (2015, 742mm/year).**
In this specific catchment and years, SFFs show equivalent or slightly higher performance than ESP at a 1-month
lead time. However, as the lead time increases, the performance of both methods tends to deteriorate. Essentially,
it indicates an underestimation in the wettest year and an overestimation in the driest year. In particular,
considerably higher performance was found in SFFs compared to ESP in the driest year (Figure 9(b)). On the
other hand, it is obvious that both methods have insufficient sharpness in forecasting flow in the wettest years for
lead times of 3 and 6 months.
Examining each ensemble member of both SFFs and ESP, we found higher variability in ESP. Furthermore, since
ESP utilizes the same weather forcings, the forecasted flows are generally similar in terms of its quantity and
patterns, regardless of the wettest and driest years. Conversely, the forecasted flow ensemble members of SFFs
show distinctive patterns for each year.
Although, these results are confined to a single catchment and specific years, this analysis is valuable in
quantitatively illustrating the forecasted flow results under dry and wet conditions and different lead times.
Furthermore, these features are generally shown in other catchments, and align with our previous findings in
section 3.4.
**4. Discussion**
**4.1 The skill of seasonal flow forecasts**

This study offers a comprehensive view of overall skill of SFFs, benchmarked to the conventional – and easier to implement - ESP method. In contrast to the majority of previous studies, which assessed the skill of SFFs at continental or national level or over large river basins, our study focuses on 12 relatively small catchments (59 - 6648 km$^2$) across South Korea.

| Lead time (months) | All seasons | | | Dry season | | | Wet season | | |
|---|---|---|---|---|---|---|---|---|---|
| | Average (2011-2020) | Dry years (2015, 2017) | Wet years (2011, 2020) | Average (2011-2020) | Dry years (2015, 2017) | Wet years (2011, 2020) | Average (2011-2020) | Dry years (2015, 2017) | Wet years (2011, 2020) |
| 1 | SFFs | SFFs | SFFs | SFFs | SFFs | SFFs | SFFs | SFFs | Equivalent |
| 2 | SFFs | SFFs | Equivalent | Equivalent | Equivalent | Equivalent | SFFs | SFFs | ESP |
| 3 | SFFs | SFFs | ESP | Equivalent | Equivalent | ESP | SFFs | SFFs | ESP |
| 4 | Equivalent | SFFs | ESP | Equivalent | SFFs | ESP | Equivalent | SFFs | ESP |
| 5 | Equivalent | SFFs | ESP | Equivalent | SFFs | ESP | Equivalent | SFFs | ESP |
| 6 | Equivalent | SFFs | ESP | Equivalent | SFFs | ESP | Equivalent | SFFs | ESP |
| 7 | Equivalent | SFFs | ESP | Equivalent | SFFs | ESP | Equivalent | SFFs | ESP |

■ : SFFs (after bias correction)   (overall skill ≥ 55%)
■ : ESP   (overall skill ≤ 45%)
■ : Equivalent   (45 < overall skill < 55%)

**Figure 10. Summary of key findings regarding the overall skill at different lead times, seasons, and years.**

Figure 10 summarizes the key findings of this study regarding the overall skill of SFFs across different seasons and years. It demonstrates that SFFs outperform ESP in almost all the cases for forecasting lead times of one month. This result is consistent with previous literature (e.g., Lucatero et al., 2018; Yossef et al., 2013). In addition, the higher skill of SFFs is also shown at lead times of 2 and 3 months in several situations as shown in Figure 10, and at even longer lead times in dry years. This is more surprising as this considerable performance of SFFs was not found in previous studies.

Similar to our study, earlier studies (Crochemore et al., 2016; Lucatero et al., 2018) have explored the skill compared with real flow observations at a catchment scale. Therefore, the comparison of their results with our findings holds interest. In brief, their results suggest that ESP remains a 'hard-to-beat' method compared to SFFs even after bias correction. Crochemore et al. (2016) showed that SFFs using bias corrected precipitation, is in equivalent level of performance with ESP up to 3 months ahead. Lucatero et al. (2018) concluded that SFFs still face difficulties in outperforming ESP, particularly at lead times longer than 1 month.

The difference of our results compared to the literature stems from a combination of several important factors. First, it is worth noting that these two previous studies were conducted at the catchment-scale, with a specific focus on Europe, namely France (Crochemore et al., 2016) and Denmark (Lucatero et al., 2018). The skill of SFFs varies according to the geographic locations, meteorological conditions of given study area, as confirmed by numerous studies (e.g., Greuell et al., 2018; Pechlivanidis et al., 2020; Yossef et al., 2013). Therefore, the skill of SFFs could also be influenced by distinct spatial and meteorological conditions between Europe and South Korea. Second, we can attribute the difference to the utilization of a more advanced seasonal weather forecasting system. Unlike previous studies which applied ECMWF system 4, our study is conducted based on ECMWF's cutting-edge forecasting system version 5. It is reported that ECMWF system 5 has many improvements compared to the previous version including the predictive skill of the El Niño Southern Oscillation (ENSO) (Johnson et al., 2019) and rainfall inter-annual variability (Köhn-Reich and Bürger, 2019). Specifically, ENSO is known to be a key driver affecting the skill of seasonal weather forecasts (Ferreira et al., 2022; Shirvani and Landman, 2015; Weisheimer & Palmer, 2014); therefore, its improvement can result in notable changes in forecasting skill. Although the relationship between seasonal weather patterns in South Korea and ENSO is not fully understood, some previous research has shown good correlations for certain regions and seasons (Lee and Julien, 2016; Noh and Ahn, 2022). While it is challenging to quantitatively evaluate the impact of system advancements in this study, given the significance of meteorological forecast in hydrological forecasts, it is highly probable that the development of the system has had a positive influence on the results. Although a few studies have analysed the skill of SFFs based on ECMWF system 5 (e.g., Peñuela et al., 2020; Ratri et al., 2023), direct comparisons with

our research were deemed difficult due to differences in spatial scale and analysis methods, such as the absence
of a comparison with ESP.
Last, the performance of the hydrological model also contributes to differences in the results. To evaluate the
impact of hydrological model to SFFs, we compared actual score (forecast performance compared to observed
flow data) with theoretical score (forecast performance compared to pseudo flow observation) and found that the
actual scores are slightly higher than theoretical scores (i.e., theoretical score shows higher performance). This
finding is consistent with previous studies, and the gap between the actual and theoretical score is highly linked
to the performance of hydrological model (Greuell et al., 2018; van Dijk, 2013). When a model's actual score
closely approximates its theoretical score, it may suggest that the model is operating at a best possible level, given
the inherent uncertainties and limitations associated with the available data and methods. Although our results
demonstrated that the theoretical score shows higher performance than actual score, their difference was generally
marginal. This close agreement between the two scores indicates that the model is well-calibrated and capable of
effectively capturing the underlying hydrological processes in those catchments.
Our findings on the impact of bias correction quantitatively showed that generally precipitation controls the
performance of SFFs, however, we also found that temperature plays a substantial role in specific seasons and
catchments. Specifically, the Hoengseong and Soyanggang catchments, located in the northernmost part of South
Korea and affected by snowfall in the Dry (winter) season (December to February), exhibit a higher temperature
contribution than precipitation for a forecasting lead time of one month during the dry season. The main reason
for this is the underestimation of temperature forecasts. Our supplementary experiments provide evidence that
using bias-corrected temperature forecasts significantly improves the performance of flow forecasts (see Figure
S4 in the supplementary material). Although the positive impact of bias correction of precipitation forecasts in
enhancing the performance of SFFs has been well-documented in numerous previous studies (Crochemore et al.,
2016; Lucatero et al., 2018; Pechlivanidis et al., 2020; Tian et al., 2018), our result demonstrates the importance
of bias correction of temperature too, at least in snow-affected catchments.
An alternative approach to bias correction has been proposed by (Lucatero et al., 2018; Yuan and Wood, 2012),
who argue that directly correcting the biases in the flow forecasts may result in better performance at a lower
computational cost. However, we tested this approach and found conflicting outcomes (Figure S9 in the
supplementary material). Therefore, caution should be exercised when directly correcting biases for flow, as this
approach may exclude the contribution of initial conditions, which is one of the most crucial factors in
hydrological modelling. In cases where the performance of hydrological model is the major source of error, bias
correction of the flow might be useful; however, if the model shows an acceptable performance, as demonstrated
in this study, incorporating bias correction for the simulated flow could add more errors.
Due to limited data availability, conducting additional validation across a larger number of extreme events is not
possible. Nevertheless, our research findings suggest a potential correlation between the overall skill and dry/wet
conditions, that should be further validated if new data become available. Specifically, in the period analysed here,
SFFs considerably outperform ESP for all lead times during the wet season in dry years. Conversely, the overall
skill during the wet season in wet years was not satisfactory. This is because the overall skill is commonly
dominated by precipitation forecasting skill, and we previously found that the skill of precipitation forecasts is the
lowest in wet years (Lee et al 2023). The systematic biases of seasonal precipitation forecasts, which tend to
underestimate (overestimate) the precipitation during the wet (dry) season, led to the consistent results in flow
forecasts. This finding also hints that SFFs hold the potential to provide valuable information for effective water
resources management during dry conditions, which is crucial for drought management.
**4.2  Limitations and directions for future research**
In this paper, we investigated the overall skill of SFFs at the catchment scale using ECMWF's seasonal weather
forecasts (system 5) with a spatial resolution of $1\times1°$. Based on our previous research, it has been demonstrated
that among four forecasting centres, ECMWF provides the most skilful seasonal precipitation forecasts (Lee et
al., 2023), thus we utilized seasonal weather forecasts datasets from ECMWF in this study. However, the skill for
other weather forcings such as temperature and PET, have not been tested across South Korea. Additionally, while
ECMWF provides seasonal weather forecasts with high resolution ($36\times36$km, approximately $0.3\times0.3°$), we
utilized publicly available low resolution data ($1\times1°$) to maintain consistency with our previous work (Lee et al.,
2023). Our additional investigation indicates that the difference in weather data between high and low resolution
is not substantial (see Figure S10 in the supplementary material). Nevertheless, prior studies suggest that the skill
of seasonal weather forecasts may vary according to factors such as region, season, and spatial resolution.
Therefore, broader research is required to determine the seasonal weather forecasts provider as well as spatial
resolution that can lead to skilful hydrological forecasts in the regions or seasons of interest.
Given the distinct climatic conditions in South Korea, it is important to acknowledge that our results may not be
applicable to other regions or countries. Therefore, further work needs to be carried out to reproduce this analysis
in different regions. To facilitate this process, two Python-based toolboxes can be useful: SEAFORM (SEAsonal
FORecasts Management) and SEAFLOW (SEAsonal FLOW forecasts). The SEAFORM toolbox, developed in
our previous study (Lee et al., 2023), offers multiple functions for manipulating seasonal weather forecast datasets
(e.g., download the datasets, time-series generation, bias correction). On the other hand, the SEAFLOW toolbox,
developed in this study, is specifically designed for the analysis of SFFs based on the modified Tank model (but
it could be useful to apply to other hydrologic models).
In terms of forecast skill, our study highlights the potential of SFFs at the catchment scale for real water resources
management. Nevertheless, it is crucial to recognize the difference between 'skill', indicating how well
hydrological forecasts mimic observed data, and 'value', referring to the practical benefits obtained from utilizing
those forecasts in real world. Previous studies have addressed this issue, showing that better skill does not always
result in higher value (Boucher et al., 2012; Chiew et al., 2003). While earlier findings suggest that the
conventional method (ESP) generally outperforms SFFs in terms of 'skill' (e.g., Lucatero et al., 2018; Yossef et
al., 2013), recent research demonstrates that, in terms of 'value,' the use of seasonal forecasts in semi-arid regions
offers significant economic benefits by mitigating hydro-energy losses in a dry year (Portele et al., 2021).
Therefore, our future research efforts should concentrate on a quantitative evaluation of the value of SFFs for
practical reservoir operations, informing decision-making in water resources management. This evaluation is of
significant importance as it directly relates to assessing the potential utilization of SFFs in practical water
management.

## 5. Conclusions

This study assessed the overall skill of SFFs across 12 catchments in South Korea using a hydrological model
forced by seasonal weather forecasts from the ECMWF (system 5). By focusing on operational reservoir
catchments with relatively small sizes, our findings showed the potential of SFFs for practical water resources
management.
The results first demonstrate that the performance of the hydrological model is crucial in flow forecasting with
the Tank model used in this study exhibiting reliable performance. Secondly, precipitation emerges as a dominant
factor influencing the performance of SFFs compared to other weather forcings, and this is more evident during
the wet season. However, temperature can also be highly important in specific seasons and catchments, and this
result highlights the significance of temperature bias correction as the flow simulation with the bias-corrected
temperature provides higher performance. Third, at catchment scale, which is more suitable for water resources
management, bias corrected SFFs have skill with respect to ESP up to 3 months ahead. Notably, the highest overall
skill during the wet season in dry years highlights the potential of SFFs to add value in drought management.
Lastly, while our research emphasizes the superior performance of SFFs at the catchment scale in South Korea, it
is important to note that outcomes may vary depending on factors such as the type of seasonal weather forecasts
system used, the study area, and the performance of the hydrological model.
As seasonal weather forecasting technologies continue to progress, it is also crucial to concurrently pursue their
application and validation in flow forecasting. We hope that our findings contribute to the ongoing validation
efforts of the skill of SFFs across various regions and, furthermore, serve as a catalyst for their practical application
in real-world water management. At the same time, our proposed workflow and the analysis package we have
developed using Python Jupyter Notebook, can offer valuable support to water managers in gaining practical
experience to utilize SFFs more effectively.
*Code and data availability*. The SEAFLOW (seasonal flow forecasts) and SEAFORM (seasonal forecast
management) Python packages are available at https://github.com/uobwatergroup/seaflow, and
https://github.com/uobwatergroup/seaform, respectively. ECMWF's seasonal weather forecasts data are available
under a range of licences from https://cds.climate.copernicus.eu/. Reservoir and flow data are made available by
the K-water and can be downloaded from https://www.water.or.kr/.
*Author contributions*. YL designed the experiments, with suggestions from the other co-authors. YL developed
the workflow and performed simulation. FP and MAR participated in repeated discussions on interpretations of
results and suggested ways forward in the analysis. AP provided YL with modelling technical support and
reviewed the manuscript.
*Competing interests*. The authors declare that they have no conflict of interest.
*Acknowledgements*. Yongshin Lee is funded through a PhD scholarship by K-water (Korea Water Resources
Corporation). Andres Peñuela is funded by the European Research Executive Agency (REA) under the
HORIZON-MSCA-2021-PF-01 grant agreement 101062258. We also thank K-water and Dr. Shinuk Kang (South
Korea, K-water Institute) for sharing data and hydrological model (modified Tank) applied in this study.

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
