# Peer review of "Skill of seasonal flow forecasts at catchment scale: an assessment across South Korea"

_EGUsphere, 2023_

## Referee Comment (RC1)

Review comments for **Skill of seasonal flow forecasts at catchment-scale: an assessment across South Korea** by Lee et al.

In this paper, the authors have conducted an evaluation of seasonal hydrological forecasts using Tank model driven by the ECMWF SEAS5 seasonal weather forecasts in various South Korean catchments. The topic is timely and brings insights to the field of hydrological forecasting. However, I have identified several areas that require attention to ensure the manuscript meets the standards of the journal.

1. The manuscript currently needs a clearer explanation of the methodologies employed, which also relates to the second point of terminology use. Some parts of method description need to be expanded. For example, the calculation of contributions from each variable needs more details. When examining the contribution of forecasted precipitation by substituting the other two variables with observations, please clarify if this was done for all ensemble members, and is it skill that you are compared on or score?

2. Consistent Use of Terminology: The paper misuses the word "*skill*" in different ways which should be sometimes specified by *performance or quality*, and this inconsistent use lead to confusion unfortunately. There seems to be also some ambiguity in the use of "*skill*" and "*score*". For instance, "theoretical skill" and "actual skill", are they referred to CRPS or CRPSS? If they are skills, could you specify which benchmark is being used to calculate them? If they are scores, then I suggest not using skill in the names. I would suggest the authors to clearly distinguish these terms and maintain consistent usage throughout the manuscript to improve clarity.

3. Abbreviation Usage: The manuscript needs a thorough review to ensure that all abbreviations are properly introduced upon their first occurrence. Additionally, to avoid redundancy, each abbreviation should only be defined once. Moreover, certain abbreviations have been assigned multiple meanings within the manuscript (for instance, CRPS at Line 183 and Line 258). This presents a significant source of confusion.

4. Actual Skill and Theoretical Skill: The authors raise an interesting point about the significance of using actual skill over theoretical skill to provide more insights for water resource management on whether to use SFFs and when. However, in the analysis, this is conducted by calculating CRPSS, with ESP as benchmark, thus the use of either actual or theoretical references does not play such a big role, as long as the benchmark is using the same reference as the forecasts. On the other hand, the information gained from theoretical skill in this paper is to validate the performance of the hydrological model by showing its proximity to the actual skill (or perhaps more appropriately, the "actual score"). In this case, it didn't really reflect the argument of providing significant information for the users.

Line by Line comments:

Line 23, "actual skill" here sounds ambiguous since there is no other information explaining this term, which might lead to misunderstanding.

Line 25, please add brief information on the methods that you use to get the conclusion that precipitation is the most important variable.

Line 57, this is the first time that ESP is mentioned (excluding abstract), therefore full explanation is needed here.

Line 77, to my knowledge, the reference Pechlivanidis et al., 2020 is not using ESP in the analysis, therefore cannot support the argument here.

Line 93, I'm a little bit suspicious on this sentence here that "only a few studies" have used SEAS5 for seasonal hydrological skill assessment. For example, the reference you mentioned before Pechlivanidis 2020 is actually using SEAS5 at higher spatial resolution.

Line 139, what is the criteria of dividing the four seasons, are they based on precipitation or flow?

Line 142, the information of annual variability is not shown in the figure but only in the text, right?

Line 143, typhoon and monsoon might not need to start with an uppercase character here.

Line 149, the abbreviation of KMA should be noted in the previous sentence when it is firstly mentioned.

Line 169, regarding SEAS5 data, here the period 1993-2020 is mentioned, but in the method part and in Figure 2, based on my understanding, the forecast period is 2011 to 2020. Please clearly specify this.

Line 181, SFFs has been mentioned many times already.

Line 183, here CRPS is referred to as skill but later it is referred to as score (Line 258).

Line 188, the plot needs to be improved. To calculate CRPS needs the forecast (either ESP or SEAS5) and the reference (either real or pseudo observation), therefore the arrows should lead from corresponding systems to the box of CRPS. However, this is not systematically shown in the plot.

Line 190, to my knowledge there is SEAS5 forecasts with higher spatial resolution that is available.

Line 205, a potential problem for linear scaling on precipitation is, it might generate very large values. Have you had any solutions to avoid this?

Line 247, as defined in Eq.4?

Line 265, what does SPFs stand for? Or maybe you mean SFFs? Otherwise please add the full name for the abbreviation.

Line 270 and Line 258, redundant information.

Line 275, Major does not need an uppercase here.

Line 275, here the CRPS of ESP is calculated using real observation as reference, it is correct?

Line 285, here comes the explanation of SPFs, but it is already mentioned many times before this.

Line 310, here I would strongly recommend to distinguish skill from score, since you have CRPSS later which are actually skills, but here these are scores.

Line 327, this part should be described in method session, and more details are needed for fully understanding.

Line 498, are these conclusions from Figure 8? Considering there are only two dry years and two wet years, the conclusion needs to be drawn carefully, otherwise it's not very scientifically valid.

Figure S1, please explain which benchmark is used here to calculate from CRPS to skill (skill-3P, skill-T).

---

## Author Comment (AC1)

**REPLY TO REFEREE#1 COMMENTS**

In this paper, the authors have conducted an evaluation of seasonal hydrological forecasts using Tank model driven by the ECMWF SEAS5 seasonal weather forecasts in various South Korean catchments. The topic is timely and brings insights to the field of hydrological forecasting. However, I have identified several areas that require attention to ensure the manuscript meets the standards of the journal.

Thank you for your valuable and insightful comments to our paper. We are committed to address them in our revision.

1. The manuscript currently needs a clearer explanation of the methodologies employed, which also relates to the second point of terminology use. Some parts of method description need to be expanded. For example, the calculation of contributions from each variable needs more details. When examining the contribution of forecasted precipitation by substituting the other two variables with observations, please clarify if this was done for all ensemble members, and is it skill that you are compared on or score?

   We thank you for this comment and agree with your point. We will provide more specific details of the methodologies used, particularly in explaining the contribution of weather variables to the performance of SFFs and clarify the skill/score issue (see next point).

2. Consistent Use of Terminology: The paper misuses the word "skill" in different ways which should be sometimes specified by performance or quality, and this inconsistent use lead to confusion unfortunately. There seems to be also some ambiguity in the use of "skill" and "score". For instance, "theoretical skill" and "actual skill", are they referred to CRPS or CRPSS? If they are skills, could you specify which benchmark is being used to calculate them? If they are scores, then I suggest not using skill in the names. I would suggest the authors to clearly distinguish these terms and maintain consistent usage throughout the manuscript to improve clarity.

   Thanks to this comment, we have recognized that our terminology might be causing confusion among readers. We will replace the term 'skill' with 'score' in sections 3.1 and 3.2, where the 'CRPS' is used. The term 'skill' is retained only in sections 3.3 and 3.4, where the overall skill (calculated using 'CRPSS') is discussed (and we will clarify that the benchmark is ESP).

3. Abbreviation Usage: The manuscript needs a thorough review to ensure that all abbreviations are properly introduced upon their first occurrence. Additionally, to avoid redundancy, each abbreviation should only be defined once. Moreover, certain abbreviations have been assigned multiple meanings within the manuscript (for instance, CRPS at Line 183 and Line 258). This presents a significant source of confusion.

   We agree with this comment. We have checked the overall usage of abbreviations and will correct them across the manuscript. Please refer to the details shown below (Line by line reply).

4. Actual Skill and Theoretical Skill: The authors raise an interesting point about the significance of using actual skill over theoretical skill to provide more insights for water resource management on whether to use SFFs and when. However, in the analysis, this is conducted by calculating CRPSS, with ESP as benchmark, thus the use of either actual or theoretical references does not play such a big role, as long as the benchmark is using the same reference as the forecasts. On the other hand, the information gained from theoretical skill in this paper is to validate the performance of the hydrological model by showing its proximity to the actual skill (or perhaps more appropriately,

the "actual score"). In this case, it didn't really reflect the argument of providing significant information for the users.

We appreciate your comment. This misunderstanding is also caused by the misuse of the term 'skill'. In this study, we use CRPSS to compute 'overall skill' by comparing the flow forecasts with real observations. In previous literature, this was often referred to as 'actual skill'. As acknowledged in the introduction, we believe that 'actual skill' (compared with real observations) would be more informative for water managers than 'theoretical skill' (compared with pseudo-observations). Additionally, to validate the performance of the hydrological model (Section 3.1), we calculated the actual and theoretical 'score' using CRPS, therefore, no benchmark is used here. To clarify this, we have modified Figure 2 (see page 4 in this document) and proposed revisions to the manuscript in line-by-line reply below.

**Line by Line comments (reply):**

Line 23, "actual skill" here sounds ambiguous since there is no other information explaining this term, which might lead to misunderstanding.

→ Agree. We will replace the term 'actual skill' with 'overall skill' with additional information explaining this term.

Line 25, please add brief information on the methods that you use to get the conclusion that precipitation is the most important variable.

→ Agree, we will add explanations of the methods.

Line 57, this is the first time that ESP is mentioned (excluding abstract), therefore full explanation is needed here.

→ Agree, we will add the full explanation of ESP.

Line 77, to my knowledge, the reference Pechlivanidis et al., 2020 is not using ESP in the analysis, therefore cannot support the argument here.

→ Agree and will remove the reference from the paragraph.

Line 93, I'm a little bit suspicious on this sentence here that "only a few studies" have used SEAS5 for seasonal hydrological skill assessment. For example, the reference you mentioned before Pechlivanidis 2020 is actually using SEAS5 at higher spatial resolution.

→ Here, we wanted to emphasize that there are not many previous studies using ECMWF SEA5 and analysing the performance of SFFs compared to ESP. To clarify our intention, we will modify the sentence.

Line 139, what is the criteria of dividing the four seasons, are they based on precipitation or flow?

→ The criteria of dividing the four seasons is based on monthly precipitation. This is intended to maintain continuity with our previous research (Lee et al., 2023) and is consistent with the general seasonal classification in South Korea. We will clarify this point.

Line 142, the information of annual variability is not shown in the figure but only in the text, right?

→ Yes, the inter-annual variability is not shown in the figure. We will include additional explanation and a reference (Lee et al., 2023) to support this.

Line 143, typhoon and monsoon might not need to start with an uppercase character here.

→ Agree, we will change them to lowercase.

Line 149, the abbreviation of KMA should be noted in the previous sentence when it is firstly mentioned.

→ Agree, we will add the abbreviation of KMA.

Line 169, regarding SEAS5 data, here the period 1993-2020 is mentioned, but in the method part and in Figure 2, based on my understanding, the forecast period is 2011 to 2020. Please clearly specify this.

→ Thank you for this comment. Our analysis focuses on the period from 2011 to 2020. However, we also analysed SEAS5 data from 1993 to 2010 to compute the bias correction factors. A detailed explanation of this process will be provided, along with Figure R1 (shown below). This figure will be added in the supplementary material.

[Figure]

**Figure R1: Observed annual precipitation (dots) from 1966 to 2020 in the 12 catchments feeding the reservoirs considered in this study. The solid line represents the mean annual precipitation over the 12 catchments. The red line represents the period for assessing the seasonal flow forecasts (2011–2020), and the blue line represents the period used to compute the bias correction factors (1993–2010). ESP ensembles are generated using observed data from 1966 to 2010.**

Line 181, SFFs has been mentioned many times already.

→ We will remove the full form of SFFs.

Line 183, here CRPS is referred to as skill but later it is referred to as score (Line 258).

→ We will use the term score consistently.

Line 188, the plot needs to be improved. To calculate CRPS needs the forecast (either ESP or SEAS5) and the reference (either real or pseudo-observation), therefore the arrows should lead from corresponding systems to the box of CRPS. However, this is not systematically shown in the plot.

→ Thank you for this suggestion. We have improved Figure 2 to clarify our methodology and the term. Please see the modified Figure 2 below.

[Figure]

**Modified Figure 2: Schematic diagram illustrating analysis method of the study.**

Line 190, to my knowledge there is SEAS5 forecasts with higher spatial resolution that is available.

→ We thank you for this comment. In this study, we have utilised the forecasts with 1×1° because we wanted to maintain consistency with our pervious study on the seasonal precipitation forecasts (Lee et al. 2023). In addition, the Copernicus Climate Data Store (CCDS) website officially states that their service has a horizontal resolution of 1×1° (https://cds.climate.copernicus.eu/cdsapp#!/dataset/seasonal-original-single-levels?tab=overview).

To clarify the relationship between the higher (36×36km, approximately 0.3×0.3°) and lower resolutions (1×1°) data, we directly contacted ECMWF. ECMWF indicated that they provide lower resolution data publicly through CCDS, and this data is generated from higher resolution data using an in-house interpolation method.

We have conducted a test to compare the forecasts with higher and lower resolution in three catchments with a large, medium, and small area, respectively. Figure R2 shown below represents the monthly P, T and PET obtained from higher and lower resolutions (before bias correction). The difference between both products is generally small, with a somewhat greater disparity in PET (orange points). However, overall, the differences were not substantial, therefore, increasing the spatial resolution may not significantly impact our results and conclusions for the catchments analysed in this study.

Additionally, we also have compared the mean P, T and PET for a single coarse grid cell (1×1°) and nine finer grid cells (36×36km) contained in the single course grid. This approach has been repeated for two different locations, see Figure R3. Again, the differences between the two datasets are small and consistent with the previous findings shown in Figure R2.

To clarify this point, we will include additional explanation in Section 4.2 (Limitations and directions for future research).

[Figure]

**Figure R2. Comparison between mean monthly weather forecasts of the high resolution product (0.3×0.3° or 36×36km, on x-axis) and low resolution (1×1°, y-axis) (first row: precipitation, second row: temperature, third row: PET) at 1- and 4-month lead times from 2011 to 2020. Analysis is repeated in three catchments: (a) large-size (Chungju), (b) medium (Yongdam) and (c) small (Buan).**

[Figure]

**Figure R3. Comparison between the mean monthly weather forecasts for a grid cell of the lower resolution (1×1°) product (vertical axis) and the average of 9 grid cells of the higher resolution (36×36km) product (horizontal axis), at different lead time (a: 1 month, b: 4 months) from 2011 to 2020. Analysis is repeated for two regions.**

Line 205, a potential problem for linear scaling on precipitation is, it might generate very large values. Have you had any solutions to avoid this?

→ Thank you for your comment. In our study, we could not find any problem generating very large values. Additionally, the linear scaling method has demonstrated its usefulness in literature (Azman et al., 2022; Crochemore et al., 2016; Shrestha et al., 2017) and South Korea (Lee et al., 2023).

Line 247, as defined in Eq.4?

→ We have corrected this typo.

Line 265, what does SPFs stand for? Or maybe you mean SFFs? Otherwise please add the full name for the abbreviation.

→ Thanks for spotting this typo. We will replace it with SFFs.

Line 270 and Line 258, redundant information.

→ Agree. We will remove this redundant information.

Line 275, Major does not need an uppercase here.

→ Agree. The words will be changed to lowercase.

Line 275, here the CRPS of ESP is calculated using real observation as reference, it is correct?

→ Yes, it is. Basically, we have computed the overall skill using real observations. To clarify this, we have modified Figure 2 (see page 4 in this document) and the sentence.

Line 285, here comes the explanation of SPFs, but it is already mentioned many times before this.

→ This has been amended in accordance with your previous comment related to Line 265.

Line 310, here I would strongly recommend to distinguish skill from score, since you have CRPSS later which are actually skills, but here these are scores.

→ Thanks for your advice. We will modify the term 'skill' to 'score' in those sentences where the CRPS is used.

Line 327, this part should be described in method session, and more details are needed for fully understanding.

→ We agree with you and have moved this part to method session and added descriptions.

Line 498, are these conclusions from Figure 8? Considering there are only two dry years and two wet years, the conclusion needs to be drawn carefully, otherwise it's not very scientifically valid.

→ We agree that having only two dry and two wet years means that we cannot draw definitive conclusions. We will revise our discussion to recognise this in our conclusions. Additionally, we have produced an additional Figure with the same analysis as Figure 8 but including data from the calibration period used to calculate bias correction factors. This extended the analysis to 5 dry years (1994, 2001, 2008, 2015, 2017) and 5 wet years (1998, 2001, 2002, 2011, 2020), respectively. (Please note that, due to the lack of observed data, here we can only use 7 catchments: Soyanaggang, Chungju, Andong, Imha, Hapcheon, Namgang, Sumjingang).

[Figure]

(a) Average over dry years ('94, '01, '08, '15, '17)    (b) Average over wet years ('98, '01, '02, '11, '20)

**Figure R4. Overall skill of bias corrected SFFs over 7 catchments averaged over (a) dry years (mean annual P < 900mm) and (b) wet years (mean annual P > 1500mm) during all seasons (black lines), dry seasons (red dashed lines) and wet seasons (blue dashed lines). Here, mean annual precipitation is averaged across the catchments and years.**

As shown in Figure R4, the results are generally consistent with Figure 8 (b, c), which is encouraging and, considering the available seasonal forecasts dataset (1993-), likely the broadest analysis that we can conduct. We will bring this discussion in the revised manuscript and acknowledged this limitation of our study.

Figure S2, please explain which benchmark is used here to calculate from CRPS to skill (skill-P, skill-T).

→ No benchmark was used; thus, we have modified Figure S2 as shown below. This figure illustrates that how we computed the contribution of each weather variable to the 'score' of SFFs (not skill).

[Figure]

**Modified Figure S2: Schematic diagram of calculating the relative scores.**

---

## Author Comment (AC2)

**REPLY TO REFEREE#2 COMMENTS**

This contribution by Lee et al. presents a performance assessment of seasonal flow forecasts generated using ECMWF SEAS5 forecasts and the Tank hydrological model, upstream 12 operational reservoirs in South Korea.

After introducing the experimental setup, the data, the hydrological model and the evaluation framework, the authors analyse the skill of seasonal flow forecasts. First the authors assess the sensitivity of the skill to the hydrological model performance, to the model inputs, namely P, PET and T, and bias correct these inputs. Then they assess the skill of seasonal flow forecasts generated based on SEAS5 with respect to the standard ESP method, distinguishing dry and wet seasons as well as dry and wet years. Lastly the authors show an example of flow forecasts in a given catchment.

Overall, this paper is well structured and written, though several typos remain in some parts of the text, which I tried to list below. The figures are relevant, informative and well presented, though the captions could sometimes be more detailed. The methodology and the analysis of results are both comprehensive and methodical. However, I list hereafter three concerns, the major one being the definition of skill in the manuscript. These are followed by a list of minor comments, mostly asking for clarifications and some reformulating.

Based on this, I recommend the paper for publication subject to major revisions as this work will be a valuable insight into the application of seasonal forecasts over South Korea with an extra focus on reservoir management.

We thank you for taking the time to review our manuscript and are grateful for the positive comments. We are committed to address them in our revision.

**General comments**

1. The term "skill" in the article is used with different meanings, and sometimes with meanings that are not consistent with the definition commonly used in the literature (see e.g. the books by Wilks, or Jolliffe and Stephenson). The skill in essence is the comparison of the performance of a forecast system with the performance of a benchmark (e.g. ESP is used here) as in Eq. 9 of the manuscript, which I refer to as "skill" in this review. This ratio ranges between -infinity and 1 (see the books by Wilks and Jolliffe and Stephenson for instance). The authors here re-employ the "overall skill" from a previous paper as the percentage of years during which the forecast system has skill with respect to the benchmark, introducing a sense of variability which is interesting. However, in the results section this becomes confusing as the authors use in turn: "actual skill" and "theoretical skill" both having values greater than 1 or 100%, "skill ratio" which is smaller than 1, "relative skill" which ranges between 0 and 100%, "overall skill" which is clearly defined, yet the term is somewhat misleading. It seems important to clarify this aspect for the paper to be understandable, to ensure scientifically sound conclusions. A non-exhaustive list of instances where this was unclear is provided in the detailed list hereafter.

    We appreciate your comments. A similar point was also raised by Referee 1. We agree that our terminology was unclear and will clarify this important point in the revised manuscript, using 'score' when referring to CRPS and 'skill' when referring to CRPSS (which uses ESP as benchmark).

2. Related to this first point, the skill thus allows the comparison of two forecasting systems. In section 3.1 it would thus seem natural to look at a skill where the numerator is the CRPS computed against pseudo-observations, and the denominator is the CRPS against real observations (or vice-versa). The result should range between -infinity and 1 if the authors

use the skill, or between 0 and 100% if they use the "overall skill". The same reasoning applies to the comparison before and after bias correction, to the experiment of skill from weather forcings, and to the comparison with the ESP (see for instance the methodologies of Crochemore et al. 2020 and Greuell et al. 2019). However, it was not entirely clear if this is what was systematically done, and I suggest clarifying this for each of the results section and in the figure captions.

We will revise the terminology across the manuscript. In addition, to clarify this issue, we have modified the schematic diagram Figure 2 (see page 6 in this document).

3. The authors did not exploit much the spatial heterogeneity in catchments, though they do mention that no correlation could be found in terms of skill with respect to the ESP (Section 3.3). I still wonder if explanations could be given regarding the Imha, Buan, and Namgang catchments which stand out in Sections 3.1 and 3.2. The Chungju catchment is also later used to illustrate forecasts. It would be useful to understand the variability that can be found between these catchments to explain differences found in the analysis. Here as well, detailed comments and suggestions are provided hereafter.

Thank you for this comment. We will incorporate further spatial characteristics of those catchments in the manuscript (Section 3.1 and 3.2). Additionally, we will include the skill of the Chungju catchments in Section 3.5. We believe that this modification will help readers with a more comprehensive understanding.

*Wilks, D. S., 2006: Statistical methods in the atmospheric sciences. Academic Press,.*
*Jolliffe, I. T., and D. B. Stephenson, 2003: Forecast Verification: A Practitioner's Guide in Atmospheric Science. John Wiley & Sons Ltd., 240 pp.*
*Crochemore, L., M.-H. Ramos, and I. G. Pechlivanidis, 2020: Can Continental Models Convey Useful Seasonal Hydrologic Information at the Catchment Scale? Water Resources Research, 56, e2019WR025700.*
*Greuell, W., W. H. P. Franssen, and R. W. A. Hutjes, 2019: Seasonal streamflow forecasts for Europe – Part 2: Sources of skill. Hydrology and Earth System Sciences, 23, 371–391.*

**Specific comments**
Abstract: There are some typos in the abstract. I suggest some modifications below but invite the authors to screen the text for typo correction. → We thank you for this comment. Once again, we will check the typos and correct them.

L14: Replace "to link" with "to generate" → We will replace it.

L15-16 "at finer scales such as catchment": A word seems to be missing → We will correct it.

L16: "generating SFFs (…) remains challenging" → We will correct it.

L19: "at catchment scale" → We will correct it.

L20: "over the last decade" → We will correct it.

L23 "actual skill": this term is not clear at this stage. Please explicit.

→ We agree with you. To clarify our methodology and goal, we will replace the term 'actual skill' with 'overall skill' in the abstract and add explanations on the 'overall skill'. We will also be consistent with the terminology when using skill or score.

L24-25: this sentence states that you compare the skill of forecasts with the ESP. It seems odd. It is rather the ESP which is used as benchmark in the skill computation, and the comparison is carried via the calculation of the skill. Please clarify. → We will clarify this sentence.

L30-31: are these "openly available"? → Yes, it is. We will change the term 'freely' to 'openly'.

L57: this is the first occurrence of ESP, please explicit the term. → We will add full explanation.

L58: "by forcing a hydrological model with historical meteorological observations" → We will correct this.

L67-68: "Some of these studies focused" → We will correct it.

L95: "did not analyse" → We will correct it.

L108: "may be considered" → We will correct it.

L110 "on assessing the actual skill and comparing it with ESP": I assume it is the actual skill of SFFs. This sentence may not work with the definition of the skill: either you compute the skill of SFFs with respect to a benchmark, and compute the skill of ESP with respect to that same benchmark, and then compare both skills, or you directly use the skill for the comparison (its intended use) and choose the CRPS of SFFs and ESP as numerator and denominator of the skill respectively. → We agree with your comment and will correct it.

Table 1: It would be informative to add Tmin and Tmax to this table, especially given that you have catchments with snow which you later discuss. → We agree with your comment and have added Tmin and Tmax in Table 1 – see modified version below.

**Modified Table 1. Mean annual (2001-2020) properties of the 12 multipurpose reservoirs (from north to south) and the catchments they drain (K-water, 2022). Here, Tmin and Tmax represent mean monthly minimum and maximum temperature averaged over 2001-2020. (P: precipitation, T: temperature, PET: potential evapotranspiration)**

| Catchment | Soyanggang | Hoengseong | Chungju | Andong | Imha | Yongdam | Buan | Sumjingang | Hapcheon | Milyang | Namgang | Jangheung |
|---|---|---|---|---|---|---|---|---|---|---|---|---|
| Area (km$^2$) | 2703 | 209 | 6648 | 1584 | 1361 | 930 | 59 | 763 | 925 | 95 | 2285 | 193 |
| P (mm) | 1220 | 1336 | 1197 | 1079 | 956 | 1317 | 1292 | 1343 | 1279 | 1375 | 1477 | 1439 |
| T (℃) | 10.8 | 10.9 | 11.1 | 11.1 | 12.2 | 11.8 | 13.5 | 12.6 | 12.8 | 14.2 | 13.5 | 13.8 |
| Mean annual T min | -4.2 (Jan.) | -4.0 (Jan.) | -3.2 (Jan.) | -3.5 (Jan.) | -1.6 (Jan.) | -2.3 (Jan.) | -0.1 (Jan.) | -1.5 (Jan.) | -0.8 (Jan.) | 1.0 (Jan.) | 0.4 (Jan.) | 1.3 (Jan.) |
| T max | 24.0 (Aug.) | 24.1 (Aug.) | 25.9 (Aug.) | 23.8 (Aug.) | 25.1 (Aug.) | 24.8 (Aug.) | 26.7 (Aug.) | 25.8 (Aug.) | 25.5 (Aug.) | 26.8 (Aug.) | 26.0 (Aug.) | 26.2 (Aug.) |
| PET (mm) | 874 | 870 | 881 | 896 | 947 | 884 | 960 | 919 | 933 | 993 | 952 | 896 |

(d,e,f): Here, instead of showing the average over all catchments, it would be interesting to represent the variability between catchments as it will later inform the variability in forecast skill. → Thank you for this suggestion. To show the inter-catchment variability in the figure, we have

[Figure]

**Modified Figure 1: Top row: mean annual (1967-2020) (a) precipitation (mm/year), (b) temperature (℃/year) and (c) PET (mm/year) across South Korea and the boundaries of the 12 reservoir catchments analysed in this study (all maps obtained by interpolating point measurements using the inverse distance weighting method). Bottom row: (d) cumulative monthly precipitation and PET, (e) mean monthly temperature and (f) cumulative monthly flow. All variables are averaged over the 12 reservoir catchments from 2001 to 2020. Box plots show the inter-catchment variability.**

In addition, in the caption: L133 "Mean monthly": isn't it rather the sum in the case of precipitation, PET and flow?  → Agree. We have revised this error. See revised caption above.

L135 "variability of each weather variable": "of each weather and hydrological variable". Is it the inter-catchment variability or the inter-annual variability?  → We have modified Figure 1 as shown above. Now, it represents the inter-catchment variability, and we will clarify this in the revised manuscript.

L149: Please introduce the abbreviation KMA here.  → We will add full explanation of KMA.

L156-160: Was the streamflow data generation done as a first step of this work? Do you make a distinction between "streamflow" and "flow" (L156)? After reading this paragraph, I was unsure whether you derived flow values (assuming flow and streamflow refer to the same variable) from measurements of river levels and a rating curve, or from a reservoir water balance, knowing

measurements of reservoir levels, inputs other than inflows, and outflows, and then a rating curve. Is it because it is the second option being carried out that reservoir evaporation is mentioned? If so, are you improving on K-water's method? → Thanks to your comment, we realised that this sentence might be unclear and could lead to misunderstandings. In this study, when we refer to 'flow data,' we specifically mean the flow to the reservoir from their upstream catchment, estimated by K-water. We meant to say that 'K-water generates flow data using the water balance equation; however, reservoir evaporation is not considered in this process.' We will change the term 'streamflow' to 'flow' and clarify this point in the manuscript.

L166: Please refer to:Johnson, S. J., and Coauthors, 2019: SEAS5: the new ECMWF seasonal forecast system. Geoscientific Model Development, 12, 1087–1117, https://doi.org/10.5194/gmd-12-1087-2019. → Agreed. We will add the reference in that sentence.

L174: Did you compute PET based on the Penman-Monteith method as mentioned L151, or did you retrieve PET forecasts directly from ECMWF? In the second case, do PET forecasts use the same method as the one used for the historical period? → Thank you for this comment. For the forecasts, we used PET data directly from ECMWF, which was computed using surface energy balance. The Penman-Monteith (PM) method requires several weather variables (such as vapor pressure, solar radiation etc.) to compute PET. However, some of these variables are not available as seasonal forecasts and therefore it was not possible to recompute the PET forecasts using the PM method.

L176: "45 ensemble members (…) were also selected from (…)" since, in my understanding, there is no generation involved. → We agree with your comment and will correct this sentence accordingly.

L177-180: Here, you first mention the construction of the ESP where each member is simulated, and then mention the parameter estimation of the hydrological model. It might be more intuitive to mention the parameter estimation before mentioning simulations. → Thank you for this suggestion. We will reorder those sentences as you suggested.

L182-184: This sentence shows the issue I have with the "skill" terminology. "The Continuous Ranked Probability Skill (CRPS) method": the CRPS is a score and not a skill, it stands for Continuous Ranked Probability Score. "method" may probably be removed. → We agree with your comment. We will use 'score' when discussing CRPS, and use 'skill' when referring to CRPSS.

Figure 2: Here the issue with the "skill" appears clearly. Skill should come from the comparison of CRPS values corresponding to two different systems. Here instead, a skill is linked to a single CRPS box, which does not make sense given the definition of skill. In addition, the method used to calculate PET could appear to clarify the point mentioned above. → Thank you for this comment. We have modified the terminology and Figure 2 shown below. As we replied above, we used PET forecasts provided by ECMWF.

[Figure]

**Modified Figure 2: Schematic diagram illustrating analysis method of the study.**

L222: "a water balance module" and "the United States" → We will correct it.

L226: "see Table S1" → We will correct it.

L234-235 "higher performance": what is meant by "performance" here? Each objective function will provide good model performances as long as we focus on the flow characteristics that the objective function focuses on. → Thank you for this comment. We intended to convey that among many possible objective functions, this objective function (Eq.4 in the manuscript) showed the best results in calibrating the Tank model as suggested by a previous study (Kang et al., 2004). To clarify this, we will make it clearer in the manuscript.

L244-247 NSE formulation: the NSE usually compares the simulation to the average of observations and not to the average of simulations. → Thank you for this correction. We will correct the typo.

L260 "the entire range of the parameter of interest": what do you mean by this? What is the parameter of interest? Do you mean "forecast range"? Please clarify. → Yes, we mean "forecast range". We will revise the sentence as you suggested.

L268: This goes against the definition of the skill and of the CRPS. The CRPS alone does not provide an estimate of the skill. → The terminology issue will be corrected across the manuscript.

L271-272 "the quality of the skill": This phrase does not make sense to me. "Quality" is what would be conveyed by the CRPS while "skill" is what is conveyed by the CRPSS. The skill is a ratio of quality/performance indices→ Thank you for this comment. We will modify this sentence.

L275: "The major reasons" → We will correct it.

L283-286 "is more skilful than the benchmark": A forecast system alone can only have skill with respect to a benchmark. Therefore, we can either say "the system gives higher performances than the benchmark" or "the system has skill with respect to …" (the two being equivalent). Similarly, the forecasting system and the benchmark cannot have the same skill. Lastly, a score of 1 does not necessarily guarantee a perfect forecast, if the benchmark is of sufficiently poor quality. → We agree with your point and will change the wording as you suggested.

L287: Usually it is not the CRPSS that is averaged due to the reasons mentioned by the authors. Rather the CRPS that is averaged over all years, and the CRPSS that is computed based on the two averaged values. → Thank you for your comment. In this study, we use a metric that we introduced in a previous study (Lee et al 2023), named 'overall skill', which measures the frequency which SFFs outperform ESP. So, the overall skill is not an averaged CRPSS but a probability that SFFs have skill with respect to the benchmark over the entire period and catchments. We will modify this sentence in the manuscript to make it clearer.

L290 "more skillful than the benchmark": please rephrase → We will rephrase this sentence considering your previous comments (L283-286).

L293 "more skillful than ESP": please rephrase. → We will rephrase this sentence considering your previous comments (L283-286).

L305: Have you identified a reason for this gap in the last three catchments? Is there a distinctive non-stationary behavior in these catchments? Or are the processes particularly hard to model with the Tank model? → Thank you for this comment. We could not find exact reason for the gap for those three catchments. However, we think it is related to the characteristics of those catchments (Imha: the driest, Namgang: the wettest, Boryung: the smallest catchment size). We will provide additional information on their characteristics.

L310 "theoretical skill measured by the mean CRPS": please rephrase. → We will change the terminology across the manuscript.

L318: It would be interesting to know why this catchment stands out. → Thank you for this comment. Imha is the driest catchment among all 12 catchments with the lowest modelling performance. Additional explanations will be included in the sentence.

Section 3.2: The results shown in Figure 5 are valuable and could help interpret the results of the comparison between SFFs and ESP if it was shown for bias adjusted variables. Figure 6 proves that the sensitivity of the skill to weather forcings is distorted due to biases. Why not show the bias adjustment first and then only the sensitivity to weather forcings so that this analysis can more easily feed the rest of the article? → Thank you for this comment. Firstly, Figure 5 shows the contribution of each weather variable to the performance of SSFs based on CRPS (i.e., there is no comparison to ESP as seen in the modified Figure 2). In addition, we aimed to demonstrate how the contribution of each variable to the performance of SFFs changes with the application of bias correction (before and

after simultaneously). Therefore, we provided the results without bias correction in the manuscript and included the bias- corrected results in the supplementary material (Figure S6).

Figure 5: There is something I do not understand in the results in Figure 5. Assuming that the relative skill represented corresponds to the overall skill resented in the Methodology section, and that the benchmark in the CRPSS is the SFF with all uncertainties (forecasts of P, T and PET). Given that precipitations are key features, replacing forecast precipitation with the observed precipitation (in skill-T and skill-PET of Figure S2) should increase the performance with respect to the benchmark (greater CRPS than that of the benchmark), and should therefore give CRPSS values greater than 0 and an overall skill greater than 50%. Here, the inverse is observed. Could you please clarify this?

→ We thank you for this comment. This misunderstanding is caused by the terminology. Since Figure 5 shows the contribution (%) of each weather variable to the performance of SFFs computed using CRPS, so there is no comparison with a benchmark. Thus, the increase or decrease in the area of each shape does not necessarily indicate an increase or decrease in performance (i.e., it represents the contribution rate (%) of each variable to the performance of SFFs). To make this clear, we modified Figure S2 as presented below.

[Figure]

**Modified Figure S2: Schematic diagram of calculating the relative performance.**

L335-337: a word may be missing. → We will revise that sentence.

L345: "which in reality" → We will correct that sentence.

L348-352: It appears clearly that the skill is degraded in some catchments, for some lead times, which could be fine if the average over years were to increase. However, this is not systematic based on Figure S3. Could the authors please comment on this and maybe add a point in the discussion

section or in the Methodology section (L199-201)? → We thank you for this suggestion. For most catchments, bias correction of weather forecasts enhances the overall skill. As you mentioned, in some catchments and lead times, the overall skill slightly deteriorates after correcting biases. It is clearly shown in the figure below (Figure R1), and we will add this figure in the supplementary material as well as additional explanations in the manuscript.

[Figure]

**Figure R1: Overall skill comparison for each catchment before (blue bar) and after bias correction (orange bar) of weather forcings (P, T and PET) at lead times of 1, 2, 4, 6 months (from top to bottom).**

L371-373: The range between 45% and 55% is somewhat subjective. I would recommend applying statistical tests instead to ensure that the full distributions of CRPS are statistically different, for instance. → Thank you for this comment. In this study, we used the concept of 'overall skill' representing the probability that SFFs outperform ESP for certain period of time for a given catchment. We believe the overall skill might provide the performance of SFFs more intuitively. The range that we used here (±5%) seems to be somewhat subjective, however, even with the statistical tests, there may still be a need for subjective choice regarding the level of confidence.

L403 "average years": Isn't Figure 8a showing results for all years, and not only average years? → We will correct it.

L408: I suggest "all years" instead of "entire years" → We will correct it.

Figure 8: Could you please indicate the number of points that is shown behind the lines? Is it the number of catchments? The number of catchments x the number of years x the number of months in

the season? Over how many points is the overall skill computed? Is it statistically representative? Would it be possible to show catchments that stand out and relate it to the analysis in Section 3.3?

→ We thank you for this comment. Here, the pale black points represent the overall skill for all seasons for each catchment and this is shown in the legend below the figure. The overall skill averaged over 2011-2020 (Figure 8a) for all seasons (black line) is computed using 10,080 data (12 catchments x 12 months x 10 years x 7 lead times). In addition, Figure S6 in the supplementary material (shown below) shows the detailed results (as overall skill rank) for each catchment and it can be related to the analysis described in Section 3.3 of the manuscript.

| | Lead time | (a) Average over 2011 – 2020 | | | | | | | (b) Average over dry years (2015, 2017) | | | | | | | (c) Average over wet years (2011, 2020) | | | | | | |
|---|---|---|---|---|---|---|---|---|---|---|---|---|---|---|---|---|---|---|---|---|---|---|
| | | 1 | 2 | 3 | 4 | 5 | 6 | 7 | 1 | 2 | 3 | 4 | 5 | 6 | 7 | 1 | 2 | 3 | 4 | 5 | 6 | 7 |
| North | Soyanggang | 7 | 7 | 6 | 4 | 4 | 4 | 7 | 11 | 8 | 10 | 9 | 9 | 11 | 10 | 8 | 11 | 6 | 4 | 4 | 3 | 2 |
| | Hoengseong | 8 | 5 | 5 | 1 | 6 | 4 | 3 | 5 | 3 | 5 | 6 | 6 | 7 | 5 | 2 | 1 | 4 | 1 | 4 | 6 | 6 |
| | Chungju | 1 | 6 | 9 | 9 | 7 | 7 | 6 | 1 | 8 | 12 | 11 | 9 | 7 | 6 | 2 | 5 | 2 | 4 | 2 | 3 | 2 |
| | Andong | 6 | 10 | 8 | 6 | 8 | 9 | 9 | 5 | 6 | 2 | 2 | 1 | 3 | 3 | 11 | 5 | 1 | 3 | 7 | 8 | 10 |
| | Imha | 2 | 8 | 11 | 11 | 11 | 10 | 11 | 5 | 7 | 10 | 9 | 11 | 7 | 9 | 5 | 5 | 8 | 9 | 8 | 7 | 6 |
| | Yongdam | 4 | 9 | 6 | 8 | 9 | 6 | 8 | 3 | 8 | 7 | 8 | 8 | 6 | 6 | 10 | 11 | 8 | 7 | 10 | 8 | 10 |
| | Buan | 8 | 12 | 12 | 12 | 12 | 12 | 12 | 3 | 12 | 8 | 11 | 12 | 12 | 12 | 11 | 10 | 12 | 12 | 10 | 11 | 8 |
| | Sumjingang | 4 | 11 | 10 | 10 | 10 | 10 | 9 | 5 | 11 | 8 | 7 | 6 | 10 | 10 | 1 | 9 | 11 | 11 | 10 | 12 | 10 |
| | Hapcheon | 3 | 2 | 3 | 3 | 3 | 3 | 4 | 1 | 3 | 2 | 1 | 1 | 1 | 1 | 5 | 3 | 8 | 9 | 8 | 8 | 9 |
| | Milyang | 11 | 3 | 4 | 7 | 4 | 7 | 5 | 9 | 2 | 2 | 2 | 1 | 4 | 6 | 2 | 1 | 2 | 6 | 4 | 3 | 2 |
| | Namgang | 8 | 1 | 1 | 4 | 2 | 2 | 2 | 10 | 1 | 1 | 2 | 4 | 2 | 2 | 8 | 3 | 4 | 7 | 3 | 1 | 2 |
| South | Jangheung | 12 | 4 | 1 | 2 | 1 | 1 | 1 | 12 | 3 | 6 | 5 | 5 | 4 | 3 | 5 | 8 | 6 | 2 | 1 | 1 | 1 |

**Figure S6: Overall skill ranks for each catchment averaged over (a) entire years (2011 to 2020), b) dry years (2015, 2017) and (c) wet years (2011, 2020) for all seasons (January to December). The catchments are arranged from the top to bottom in order of their location from the northernmost (Soyanggang) to the southernmost (Jangheung). The three most (least) skilful reservoirs are highlighted in yellow (pink) colour.**

Section 3.5: It would help the reader to know the CRPS or skill obtained in this catchment. In addition, an underestimation in wet years and an overestimation in dry years are observed. Could the authors comment on this? → Thank you for this suggestion. We will add explanations describing the overall skill of the Chungju catchment and the features of underestimation and overestimation in Section 3.5.

L424: On Figure 9, it seems obvious for the 1-month cumulative flows, but not necessarily for the other time periods. → We agree with you and will revise the sentence to clarify this.

L462: This point is very relevant. It would be interesting for readers that are not familiar with the area whether ENSO is a good predictor in South Korea. → Thank you for this comment. To enhance this point, we will add explanations on the significance of ENSO in the skill of seasonal forecasts, along with insights into the connection between regional weather patterns and ENSO in South Korea, as supported by previous studies.

L496: "useful; however" → We will correct it.

L507-508 "we investigated the skill of seasonal weather forecasts": in my understanding, all analyses focus on the skill of seasonal flow forecasts. → We agree with you and will modify this sentence to convey our message more clearly.

L511: "have not been tested" → We will correct it.

L511 "more broadly research": I am no sure this is correct. Please consider rephrasing. → We will modify this sentence (to 'broader research').

---

## Author Response (AR1)

**Author response**

We thank the editor and referees for their careful reading and helpful comments. Our reply is given below. The page and line numbers correspond to the modifications done on the revised manuscript.

**Reviewer 1**

Line 23, "actual skill" here sounds ambiguous since there is no other information explaining this term, which might lead to misunderstanding. → We have replaced the term 'actual skill' with 'overall skill' with additional information explaining this term. (Page 1. Line 25)

Line 25, please add brief information on the methods that you use to get the conclusion that precipitation is the most important variable. → We have added explanations of the methods. (Page 1. Line 23-24)

Line 57, this is the first time that ESP is mentioned (excluding abstract), therefore full explanation is needed here. → We have added the full explanation of ESP. (Page 2. Line 58-59)

Line 77, to my knowledge, the reference Pechlivanidis et al., 2020 is not using ESP in the analysis, therefore cannot support the argument here. → We removed the reference from the paragraph. (Page 2. Line 77)

Line 93, I'm a little bit suspicious on this sentence here that "only a few studies" have used SEAS5 for seasonal hydrological skill assessment. For example, the reference you mentioned before Pechlivanidis 2020 is actually using SEAS5 at higher spatial resolution. → We wanted to emphasize that there are not many previous studies using ECMWF SEA5 and analysing the performance of SFFs compared to ESP. To clarify our intention, we modified the sentence. (Page 2. Line 93-95)

Line 139, what is the criteria of dividing the four seasons, are they based on precipitation or flow? → The criteria of dividing the four seasons is based on monthly precipitation. This is intended to maintain continuity with our previous research (Lee et al., 2023) and is consistent with the general seasonal classification in South Korea. (Page 4. Line 141-143)

Line 142, the information of annual variability is not shown in the figure but only in the text, right? → Yes. We included additional explanation and a reference (Lee et al., 2023) to support this. (Page 4. Line 145-146)

Line 143, typhoon and monsoon might not need to start with an uppercase character here. → It has been replaced to lowercase. (Page 4. Line 146)

Line 149, the abbreviation of KMA should be noted in the previous sentence when it is firstly mentioned. → We have added the abbreviation of KMA. (Page 4. Line 152)

Line 169, regarding SEAS5 data, here the period 1993-2020 is mentioned, but in the method part and in Figure 2, based on my understanding, the forecast period is 2011 to 2020. Please clearly specify this. → Our analysis focuses on the period from 2011 to 2020. However, we also analysed SEAS5 data from 1993 to 2010 to compute the bias correction factors. A detailed explanation of this process is provided as a Figure in the supplementary material (Figure S1). (Page 4. Line 173-174)

Line 181, SFFs has been mentioned many times already. → We have removed the full form of SFFs. (Page 5. Line 181)

Line 183, here CRPS is referred to as skill but later it is referred to as score (Line 258). → The issue of terminology for score and skill has been modified across revised manuscripts.

Line 188, the plot needs to be improved. To calculate CRPS needs the forecast (either ESP or SEAS5) and the reference (either real or pseudo-observation), therefore the arrows should lead from corresponding systems to the box of CRPS. However, this is not systematically shown in the plot. → We have improved Figure 2 to clarify our methodology and the term. (Page 5. Line 189)

Line 190, to my knowledge there is SEAS5 forecasts with higher spatial resolution that is available. → To clarify this issue, we have conducted a test to compare the forecasts with higher and lower resolution in three catchments. We have included additional explanation in Section 4.2 (Page 15. Line 544-551) and added the comparison result in the supplementary material (Figure S10).

Line 205, a potential problem for linear scaling on precipitation is, it might generate very large values. Have you had any solutions to avoid this? → In our study, we could not find any problem generating very large values. We also added references supporting our choice. (Page 6. Line 215-216)

Line 247, as defined in Eq.4? → We have corrected this typo. (Page 7. Line 262)

Line 265, what does SPFs stand for? Or maybe you mean SFFs? Otherwise please add the full name for the abbreviation. → We have corrected this typo. (Page 7. Line 280)

Line 270 and Line 258, redundant information. → We have removed the redundant sentence. (Page 8. Line 286)

Line 275, Major does not need an uppercase here. → We have removed the wording.

Line 275, here the CRPS of ESP is calculated using real observation as reference, it is correct? → Yes, it is. This is now clearly shown in Figure 2. (Page 5. Line 189)

Line 285, here comes the explanation of SPFs, but it is already mentioned many times before this. → SPF is a typo, we have amended this across the manuscript.

Line 310, here I would strongly recommend to distinguish skill from score, since you have CRPSS later which are actually skills, but here these are scores. → We have modified the term 'skill' to 'score' in those sentences where the CRPS is used.

Line 327, this part should be described in method session, and more details are needed for fully understanding. → We have moved this part to method section and added descriptions. (Page 5. Line 194-201)

Line 498, are these conclusions from Figure 8? Considering there are only two dry years and two wet years, the conclusion needs to be drawn carefully, otherwise it's not very scientifically valid. → We agree that having only two dry and two wet years means that we cannot draw definitive conclusions. We have revised our discussion to recognise this in our conclusions. Additionally, we have produced an additional Figure with the same analysis as Figure 8 but including data from the calibration period used to calculate bias correction factors. This extended the analysis to 5 dry years (1994, 2001, 2008, 2015, 2017) and 5 wet years (1998, 2001, 2002, 2011, 2020), respectively. (Please note that, due to

the lack of observed data, here we can only use 7 catchments: Soyanaggang, Chungju, Andong, Imha, Hapcheon, Namgang, Sumjingang).

[Figure]

**Figure R1. Overall skill of bias corrected SFFs over 7 catchments averaged over (a) dry years (mean annual P < 900mm) and (b) wet years (mean annual P > 1500mm) during all seasons (black lines), dry seasons (red dashed lines) and wet seasons (blue dashed lines). Here, mean annual precipitation is averaged across the catchments and years.**

  As shown in Figure R1, the results are generally consistent with Figure 8 (b, c), which is encouraging and, considering the available seasonal forecasts dataset (1993-), likely the broadest analysis that we can conduct. We also included additional information on this issue in the Discussion section. (Page 15. Line 529-531)

Figure S2, please explain which benchmark is used here to calculate from CRPS to skill (skill-P, skill-T).→ No benchmark was used here. To clarify this, we have modified Figure S2 (supplementary material).

**Reviewer 2**

L14: Replace "to link" with "to generate" → We have replaced it. (Page 1. Line 14)

L15-16 "at finer scales such as catchment": A word seems to be missing → We have corrected it. (Page 1. Line 15-16)

L16: "generating SFFs (…) remains challenging" → We have corrected it. (Page 1. Line 15)

L19: "at catchment scale" → We have corrected it. (Page 1. Line 19)

L20: "over the last decade" → We have corrected it. (Page 1. Line 20)

L23 "actual skill": this term is not clear at this stage. Please explicit.

→ We agree with you. To clarify our methodology and goal, we have replaced the term 'actual skill' with 'overall skill' in the abstract and add explanations on the 'overall skill'. (Page 1. Line 25)

L24-25: this sentence states that you compare the skill of forecasts with the ESP. It seems odd. It is rather the ESP which is used as benchmark in the skill computation, and the comparison is carried via the calculation of the skill. Please clarify. → We have clarified this sentence. (Page 1. Line 25)

L30-31: are these "openly available"? → Yes, it is. We used 'open-source Python package'. (Page 1. Line 32)

L57: this is the first occurrence of ESP, please explicit the term. → We have added the full explanation of ESP. (Page 2. Line 58-59)

L58: "by forcing a hydrological model with historical meteorological observations" → We have corrected this. (Page 2. Line 60)

L67-68: "Some of these studies focused" → We have corrected this typo. (Page 2. Line 68)

L95: "did not analyse" → We have corrected it. (Page 2. Line 95)

L108: "may be considered" → We have corrected it. (Page 2. Line 108)

L110 "on assessing the actual skill and comparing it with ESP": I assume it is the actual skill of SFFs. This sentence may not work with the definition of the skill: either you compute the skill of SFFs with respect to a benchmark, and compute the skill of ESP with respect to that same benchmark, and then compare both skills, or you directly use the skill for the comparison (its intended use) and choose the CRPS of SFFs and ESP as numerator and denominator of the skill respectively. → We have corrected the sentence. (Page 2. Line 110)

Table 1: It would be informative to add Tmin and Tmax to this table, especially given that you have

catchments with snow which you later discuss. → We have added Tmin and Tmax in Table 1. (Page 4. Line 137)

(d,e,f): Here, instead of showing the average over all catchments, it would be interesting to represent the variability between catchments as it will later inform the variability in forecast skill.
→ We have added box plots in Figure 1 (d, e, f). (Page 3. Line 128)

In addition, in the caption: L133 "Mean monthly": isn't it rather the sum in the case of precipitation, PET and flow? → We have revised this error. (Page 3. Line 129-133)

L135 "variability of each weather variable": "of each weather and hydrological variable". Is it the inter-catchment variability or the inter-annual variability? → To clarify this issue, we have modified the caption. (Page 3. Line 129-133)

L149: Please introduce the abbreviation KMA here. → We have added full explanation of KMA. (Page 4. Line 152)

L156-160: Was the streamflow data generation done as a first step of this work? Do you make a distinction between "streamflow" and "flow" (L156)? After reading this paragraph, I was unsure whether you derived flow values (assuming flow and streamflow refer to the same variable) from measurements of river levels and a rating curve, or from a reservoir water balance, knowing measurements of reservoir levels, inputs other than inflows, and outflows, and then a rating curve. Is it because it is the second option being carried out that reservoir evaporation is mentioned? If so, are you improving on K-water's method? → We have changed the term 'streamflow' to 'flow' and clarify this point. (Page 4. Line 160)

L166: Please refer to:Johnson, S. J., and Coauthors, 2019: SEAS5: the new ECMWF seasonal forecast system. Geoscientific Model Development, 12, 1087–1117, https://doi.org/10.5194/gmd-12-1087-2019. → We have added the reference in that sentence. (Page 4. Line 169)

L174: Did you compute PET based on the Penman-Monteith method as mentioned L151, or did you retrieve PET forecasts directly from ECMWF? In the second case, do PET forecasts use the same method as the one used for the historical period? → For the forecasts, we used PET data directly from ECMWF, which was computed using surface energy balance. The Penman-Monteith (PM) method requires several weather variables (such as vapor pressure, solar radiation etc.) to compute PET. However, some of these variables are not available as seasonal forecasts and therefore it was not possible to recompute the PET forecasts using the PM method.

L176: "45 ensemble members (…) were also selected from (…)" since, in my understanding, there is no generation involved. → We have corrected this sentence. (Page 5. Line 182-183)

L177-180: Here, you first mention the construction of the ESP where each member is simulated, and then mention the parameter estimation of the hydrological model. It might be more intuitive to mention the parameter estimation before mentioning simulations. → We moved those sentences as you suggested. (Page 5. Line 179-185)

L182-184: This sentence shows the issue I have with the "skill" terminology. "The Continuous Ranked Probability Skill (CRPS) method": the CRPS is a score and not a skill, it stands for Continuous Ranked Probability Score. "method" may probably be removed. → We used the term 'score' when discussing CRPS, and use 'skill' when referring to CRPSS.

Figure 2: Here the issue with the "skill" appears clearly. Skill should come from the comparison of CRPS values corresponding to two different systems. Here instead, a skill is linked to a single CRPS box, which does not make sense given the definition of skill. In addition, the method used to calculate PET could appear to clarify the point mentioned above. → We have modified the general terminology across the revised manuscript as well as Figure 2. (Page 5. Line 189)

L222: "a water balance module" and "the United States" → We have corrected them. (Page 6. Line 237)

L226: "see Table S1" → We have corrected it. (Page 7. Line 241)

L234-235 "higher performance": what is meant by "performance" here? Each objective function will provide good model performances as long as we focus on the flow characteristics that the objective function focuses on. → To clarify this, we have modified the sentence. (Page 7. Line 249)

L244-247 NSE formulation: the NSE usually compares the simulation to the average of observations and not to the average of simulations. → We have corrected the typo. (Page 7. Line 259-262)

L260 "the entire range of the parameter of interest": what do you mean by this? What is the parameter of interest? Do you mean "forecast range"? Please clarify. → We have revised the sentence as you suggested. (Page 7. Line 275)

L268: This goes against the definition of the skill and of the CRPS. The CRPS alone does not provide an estimate of the skill. → The terminology issue has been corrected across the manuscript.

L271-272 "the quality of the skill": This phrase does not make sense to me. "Quality" is what would be conveyed by the CRPS while "skill" is what is conveyed by the CRPSS. The skill is a ratio of quality/performance indices → We have removed this sentence.

L275: "The major reasons" → We have removed this expression.

L283-286 "is more skilful than the benchmark": A forecast system alone can only have skill with respect to a benchmark. Therefore, we can either say "the system gives higher performances than the benchmark" or "the system has skill with respect to …" (the two being equivalent). Similarly, the forecasting system and the benchmark cannot have the same skill. Lastly, a score of 1 does not necessarily guarantee a perfect forecast, if the benchmark is of sufficiently poor quality. → We agree with your point and have changed the wording as you suggested across the revised manuscript.

L287: Usually it is not the CRPSS that is averaged due to the reasons mentioned by the authors. Rather the CRPS that is averaged over all years, and the CRPSS that is computed based on the two

averaged values. → In this study, we use a metric that we introduced in a previous study (Lee et al 2023), named 'overall skill', which measures the frequency which SFFs outperform ESP. So, the overall skill is not an averaged CRPSS but a probability that SFFs have skill with respect to the benchmark over the entire period and catchments.

L290 "more skillful than the benchmark": please rephrase → We have rephrased these sentences across the revised manuscript considering your previous comments (L283-286).

L293 "more skillful than ESP": please rephrase. → We have rephrased these sentences across the revised manuscript considering your previous comments (L283-286).

L305: Have you identified a reason for this gap in the last three catchments? Is there a distinctive non-stationary behavior in these catchments? Or are the processes particularly hard to model with the Tank model? → We could not find exact reason for the gap for those three catchments. However, we think it is related to the characteristics of those catchments (Imha: the driest, Namgang: the wettest, Boryung: the smallest catchment size). We have provided additional information on their characteristics. (Page 9. Line 320-323)

L310 "theoretical skill measured by the mean CRPS": please rephrase. → We have changed the terminology across the manuscript.

L318: It would be interesting to know why this catchment stands out. → Thank you for this comment. Imha is the driest catchment among all 12 catchments with the lowest modelling performance. Additional explanations are included in the sentence. (Page 9. Line 334-336)

Section 3.2: The results shown in Figure 5 are valuable and could help interpret the results of the comparison between SFFs and ESP if it was shown for bias adjusted variables. Figure 6 proves that the sensitivity of the skill to weather forcings is distorted due to biases. Why not show the bias adjustment first and then only the sensitivity to weather forcings so that this analysis can more easily feed the rest of the article? → Figure 5 shows the contribution of each weather variable to the performance of SSFs based on CRPS (i.e., there is no comparison to ESP as seen in the modified Figure 2). In addition, we aimed to demonstrate how the contribution of each variable to the performance of SFFs changes with the application of bias correction (before and after simultaneously). Therefore, we provided the results without bias correction in the manuscript and included the bias-corrected results in the supplementary material (Figure S7).

Figure 5: There is something I do not understand in the results in Figure 5. Assuming that the relative skill represented corresponds to the overall skill resented in the Methodology section, and that the benchmark in the CRPSS is the SFF with all uncertainties (forecasts of P, T and PET). Given that precipitations are key features, replacing forecast precipitation with the observed precipitation (in skill-T and skill-PET of Figure S2) should increase the performance with respect to the benchmark (greater CRPS than that of the benchmark), and should therefore give CRPSS values greater than 0 and an overall skill greater than 50%. Here, the inverse is observed. Could you please clarify this?

→ This misunderstanding is caused by the terminology. Since Figure 5 shows the contribution (%) of

each weather variable to the performance of SFFs computed using CRPS, so there is no comparison with a benchmark. Thus, the increase or decrease in the area of each shape does not necessarily indicate an increase or decrease in performance (i.e., it represents the contribution rate (%) of each variable to the performance of SFFs). To make this clear, we modified Figure S2 in supplementary material.

L335-337: a word may be missing. → We have revised that sentence. (Page 9. Line 347-350)

L345: "which in reality" → We have corrected the sentence. (Page 9. Line 358)

L348-352: It appears clearly that the skill is degraded in some catchments, for some lead times, which could be fine if the average over years were to increase. However, this is not systematic based on Figure S3. Could the authors please comment on this and maybe add a point in the discussion section or in the Methodology section (L199-201)? → For most catchments, bias correction of weather forecasts enhances the overall skill. As you mentioned, in some catchments and lead times, the overall skill slightly deteriorates after correcting biases. We have added a figure supports this in the supplementary material (Figure S6) as well as additional explanations in the manuscript. (Page 10. Line 366-371)

L371-373: The range between 45% and 55% is somewhat subjective. I would recommend applying statistical tests instead to ensure that the full distributions of CRPS are statistically different, for instance. → In this study, we used the concept of 'overall skill' representing the probability that SFFs outperform ESP for certain period of time for a given catchment. We believe the overall skill might provide the performance of SFFs more intuitively. The range that we used here (±5%) seems to be somewhat subjective, however, even with the statistical tests, there may still be a need for subjective choice regarding the level of confidence. We have provided additional explanations for the reason in the modified manuscript. (Page 11. Line 389-393)

L403 "average years": Isn't Figure 8a showing results for all years, and not only average years? → We have corrected it. (Page 12. Line 428)

L408: I suggest "all years" instead of "entire years" → We have corrected it as you advised. (Page 12. Line 417)

Figure 8: Could you please indicate the number of points that is shown behind the lines? Is it the number of catchments? The number of catchments x the number of years x the number of months in the season? Over how many points is the overall skill computed? Is it statistically representative? Would it be possible to show catchments that stand out and relate it to the analysis in Section 3.3?

→ Here, the pale black points represent the overall skill for all seasons for each catchment and this is shown in the legend below the figure. The overall skill averaged over 2011-2020 (Figure 8a) for all seasons (black line) is computed using 10,080 data (12 catchments x 12 months x 10 years x 7 lead times). In addition, Figure S8 in the supplementary material shows the detailed results (as overall skill rank) for each catchment and it can be related to the analysis described in Section 3.3 of the manuscript.

Section 3.5: It would help the reader to know the CRPS or skill obtained in this catchment. In addition, an underestimation in wet years and an overestimation in dry years are observed. Could the authors comment on this? → We have added explanations describing the overall skill of the Chungju catchment and the features of underestimation and overestimation. (Page 12. Line 435-437, Page 13. Line 445-446)

L424: On Figure 9, it seems obvious for the 1-month cumulative flows, but not necessarily for the other time periods. → We have revised the sentence to clarify this. (Page 13. Line 448-449)

L462: This point is very relevant. It would be interesting for readers that are not familiar with the area whether ENSO is a good predictor in South Korea. → To enhance this point, we have added explanations on the significance of ENSO in the skill of seasonal forecasts, along with insights into the connection between regional weather patterns and ENSO in South Korea, as supported by previous studies. (Page 14. Line 488-493)

L496: "useful; however" → We have corrected it. (Page 15. Line 527)

L507-508 "we investigated the skill of seasonal weather forecasts": in my understanding, all analyses focus on the skill of seasonal flow forecasts. → We have modified this sentence to convey our message more clearly. (Page 15. Line 540-541)

L511: "have not been tested" → We have corrected it. (Page 15. Line 544)

L511 "more broadly research": I am no sure this is correct. Please consider rephrasing. → We have modified this sentence. (Page 15. Line 550)

---

## Author Response (AR2)

**Author response**

We thank the editor and referees for their careful reviewing and helpful comments. Our reply is given below. The page and line numbers correspond to the modifications done on the revised manuscript.

**Editor**

L54: lack of forecast performance → This has been corrected. (Page 2. Line 54)

L68: Numerous studies have been conducted → This has been corrected. (Page 2. Line 68)

L73: specific regions → This has been corrected. (Page 2. Line 73)

L82 wether to use SFFs, and when. → This has been corrected. (Page 2. Line 82)

L84: check spelling of reference for van Dijk → We have corrected this typo. (Page 2. Line 84)

L85: Mention is made of the coarse resolution of seasonal forecasts, suggesting these are 1x1 degrees. I would agree that this is the case for seasonal forecasts as made available through e.g. the CDS from which the forecasts have been download. It may be good to note that the original resolution of these forecasts is in fact on the order of 0.4x0.4 degrees, which have been resampled on the CDS. Please also see the documentation and references for the SEAS5 forecasts.
→ We thank you for this comment. In the Introduction section, we aim to present previous findings on the overall impact of bias correction. The details regarding the resolution of ECMWF's seasonal weather forecasts used in this study, are described in Section 2.1.2 (Data) and Section 4.2 (Limitations and directions for future research).

L104-L106: I appreciate that it is relevant to study the skill of seasonal forecasts in South Korea. However, I would consider the remark that precipitation is correlated to flow to be somewhat trivial as this applies to most catchments in the world.
→ We agree with your point, so this sentence has been removed.

L113: and also → This has been corrected. (Page 3. Line 113)

L123: rephrase this as: the catchments upstreamt of 12 multi-purpose reservoir
→ We agree with you and have rephrased the sentence as suggested. (Page 3. Line 122)

L125: the locations → This has been corrected. (Page 3. Line 125)

L138: please drop "changes in" as this is confusing - monthly precipitation and PET is sufficient.
→ We agree with the editor's comment, and we dropped 'changes in'. (Page 4. Line 137)

L141: vary → This has been corrected. (Page 4. Line 140)

L141: divide the year → This has been corrected. (Page 4. Line 140)

L150: simulate flow → This has been corrected. (Page 4. Line 149)

L151/L153: "produced" and "generated" are strange verbs to use in this context. Consider using "from"
→ We have replaced them to 'from'. (Page 4. Line 150-153)

L156: provide reference to FAO.
→ Thank you for this comment. We have added the reference. (Page 4. Line 155)

L160: I presume you mean hear the inflow to the reservoirs. May be good call this "daily inflow data" to make that clear. → We agree and have corrected it. (Page 4. Line 159)

L161: Instead of water supplies - I would describe this as releases from the reservoir - as that is what I presume you mean → We agree and have corrected it. (Page 4. Line 161)

L191: we analyse (note: given that this is presented in the current paper it should be present tense)
→ We agree and have corrected it. (Page 5. Line 190)

L228: original forecast datasets → This has been corrected. (Page 6. Line 227)

L229: each seasonal weather forecast → This has been corrected. (Page 6. Line 228)

L233: soil moisture structures → This has been corrected. (Page 6. Line 232)

L234: it is not clear what the comparison, so rather than "higher" just mention "good"
→ We agree and have corrected it. (Page 6. Line 233)

L281: H(x) is often referred to as the Heavside Function, rather than the Indicator function - hence the Heavside
→ Thank you for this advice. We have replaced it to 'Heaviside function'. (Page 7. Line 280)

L282: members would exactly match the observations, and CRPS would equal 0.
→ This has been corrected. (Page 7. Line 281)

L283: a lower performance → This has been corrected. (Page 7. Line 282)

L287: I would say it compares to a benchmark forecast, as it is not a comparison to the method, but rather an application of the method → We agree and have corrected it. (Page 7. Line 286)

L293: has a lower performance than the benchmark → This has been corrected. (Page 8. Line 292)

L299: Low or no skill → This has been corrected. (Page 8. Line 297-298)

L338-L360: In this section and ensuing sections the terminology refering to the variables (Temperature, Precipiation and PET) that are used as forcing either from the observed dataset or from the forecast dataset. Various terms are used; weather forecasting forecast, weather forecats, weather forcing, etc. This makes the section confusing to read. This should be clarified and a consistent terminology used.
→ We thank you for this comment. To maintain consistency in the use of terminology throughout this paragraph, we have adjusted these expressions to 'weather forcing' (Page 9, Line 337-359).

L376: which leads → This has been corrected. (Page 10. Line 375)

L376: correct partitioning → This has been corrected. (Page 10. Line 375)

L392: should this read - to assist analysis? I am not sure who "analysts" refers to → We agree with your comment. We have replaced it to analysis. (Page 11. Line 391)

L446: should this read: wettest years and driest years? Or is this only the case in these two single years?
→ Thank you for this comment. Yes, it is the case in two single years, but they are the wettest and driest years in history in Chungju reservoir. To avoid confusion, we have added the specific year next to wettest and driest year. (Page 13. Line 445)

L475: has an equivalent level of performance → This has been corrected. (Page 14. Line 474-475)

L481: geographic locations → This has been corrected. (Page 14. Line 480)

L493-495: the sentence starting with "While" is somewhat confusing, please rephrase
→ We agree with you. We have modified the sentence. (Page 14. Line 492-494)

L500: the actual score → This has been corrected. (Page 15. Line 499)

L501: the theoretical score → This has been corrected. (Page 15. Line 500)

L563: the real world → This has been corrected. (Page 16. Line 563)

**Referee 1**

Line 145, is it "inter-annual" or "intra-annual" that is referred to?
→ This is additional information regarding the inter-annual climatic characteristic in South Korea. We have modified and relocated the sentence to improve its clarity (Page 4. Line 144-147)

**Referee 2**

L55: "real-world applications" → This has been corrected. (Page 2. Line 55)

L69 and L81: I would suggest introducing specific terminologies chosen in this paper ("actual skill", "theoretical skill") in the Methodology section instead, because the terminologies are not clearly indicated as coming from the literature, and because it is easier to identify the term with the help of Figure 2. In addition, here you still refer to skill, and in particular "theoretical skill", but you calculate 'actual scores', 'theoretical scores' and a skill which could be called "actual skill" but no "theoretical skill".
→ Thank you for your comment. We believe that separating the two skills in the introduction would indeed help in clarifying their respective meanings. Additionally, the following literature reviews are subject to each skill.

L107: "of various sizes" → This has been corrected. (Page 2. Line 106)

L110: I would suggest "long-term probability" → This has been corrected. (Page 2. Line 109)

L110: Here as well, the notion of "overall skill" might be too specific for the introduction. You could simply refer to "skill" here and detail the terminologies (actual, theoretical, actual) in the Methodology section.
→ Thank you for this comment. To prevent confusion with the term 'skill' in following sections, it would be more appropriate to use 'overall skill' from the introduction.

L135: Just to be sure I understand correctly: for each month, you first take the minimum (or maximum), and then average the values obtained for a given month over all years. Is that correct?
→ Yes, it is correct.

L192: Consider adding "calculated using flow observations" after "actual score" to
→ We agree with you and have modified it. (Page 5. Line 191)

L194: Here you ignore all errors from the hydrological model, and not only the ones due to the structure. Consider expanding. → This has been corrected. (Page 5. Line 193)

L195: "influences" → This has been corrected. (Page 5. Line 194)

L249-250: In my opinion, this statement still requires some clarification. Results may be superior but surely for certain flow characteristics. This objective function should be good to fix the bias, but will, for instance, not focus on extreme values.
→ Based on the previous research (Kang et al., 2004), this objective function generally showed good performance, not for specific conditions. We have modified the sentence. (Page 7. Line 248)

L260-261: This is not a critical issue, but here you are considering two metrics that resemble bias (assessing overestimation and underestimation), and that are redundant. → Although they can both be used to assess overestimation and underestimation, they have distinct physical meanings.

L283: "indicates a lower performance" → This has been corrected. (Page 7. Line 282)

L299: "the presence of few extremely negative values" → This has been corrected. (Page 8. Line 298)

L320-323: Here, a factor that may better explain the loss from calibration and validation is rather a temporal one, such as a trend or inhomogeneity. Could human influences in these three catchments cause this loss in performance?
→ Thank you for this comment. In selecting target reservoirs, we ruled out all the reservoirs with human influence, such as artificial inflow. (Page 3. Line 122-124)

L329 "based on accumulated monthly mean flow": I am not sure I understand this. Based on the Methodology section it seemed you were considering monthly mean flow. Do I understand correctly here that you still take the monthly average and then accumulate it over lead months? If you refer to the variable shown in Figure 9, values do not seem to be averaged monthly. Regarding the decrease in score values with the lead time, this is expected due to the loss of information from initial conditions, both in the weather and hydrological systems.
→ In calculating monthly CRPS, we used accumulated flow for a given lead time. The value of CRPS for each reservoir, shown in Figure 4, represents the mean monthly CRPS for the reservoir. However, Figure 9 illustrates cumulated daily simulated flow for specific lead times, rather than scores or skills, so it is not averaged.

L335: "the driest" → This has been corrected. (Page 9. Line 334)

L347-350: Please consider rephrasing this sentence. The subject of "forced" is not fully clear.
→ We have rephrased the sentence. (Page 9. Line 346-348)

L419-420: This phrasing is confusing. I suggest rephrasing simply to "The pale black points represent the overall skill in each catchment." → This has been corrected. (Page 12. Line 487-419)

L421: This does not seem true for the dry season, since SFFs outperform ESP only (maybe) in the first lead month. → We agree with you and have modified it. (Page 12. Line 420-421)

L422: "SFFs are" → This has been corrected. (Page 12. Line 421)

L431: "in the Southern region" → This has been corrected. (Page 12. Line 430)

L444-445: This sentence now seems redundant with the one added. Please consider removing.
→ They have different meaning. While added sentence (Line 434-435) explains the general tendency of 'overall skill' across entire period, this sentence (Line 443-444) refers to the performance of flow forecast in specific year and season, explaining Figure 9.

L446: I suggest nuancing. For instance, for the wettest year, there is underestimation at the scale of the season but overestimation in the first lead month.
→ It is difficult to agree that 'there is overestimation in the first lead month'. But we slightly modified the sentence as you suggested. (Page 13. Line 445-446)

L448-449: It seems to depend on the horizon you look at. If ensembles were less sharp, they might have had a chance at including the observation in their range (Jul-Oct).
→ We agree with you that the term 'sharpness' might raise confusion; therefore, we have replaced it to performance. (Page 13. Line 447)